


# Surface representation impacts on turbulent heat fluxes in WRF (v.4.1.3)

Román-Cascón Carlos[1,2], Lothon Marie[2], Lohou Fabienne[2], Hartogensis Oscar[3], Vila-Guerau de Arellano Jordi[3], Pino David[4], Yagüe Carlos[5], and Pardyjak Eric R.[6]

[1]Centre National d'Études Spatiales (CNES), 31400 Toulouse, France.
[2]Laboratorie d'Aerologie, CNRS, Université de Toulouse, 31400 Toulouse, France.
[3]Meteorology and Air Quality Section, Wageningen University, Wageningen, Netherlands.
[4]Department of Physics, Universitat Politècnica de Catalunya-BarcelonaTech, 08034 Barcelona, Spain.
[5]Departamento de Física de la Tierra y Astrofísica. Universidad Complutense de Madrid, 28040 Madrid, Spain.
[6]Department of Mechanical Engineering, University of Utah, Salt Lake City, UT, USA

**Correspondence:** Carlos Román Cascón (carlosromancascon@ucm.es)

**Abstract.** The water and energy transfers at the interface between the Earth's surface and the atmosphere should be correctly simulated in numerical weather and climate models. This implies the need for a realistic and accurate representation of land cover (LC), including appropriate parameters for each vegetation type. In some cases, the lack of information and crude representation of the surface leads to errors in the simulation of soil and atmospheric variables. This work investigates the

ability of the Weather Research and Forecasting (WRF) model to simulate surface heat fluxes in a heterogeneous area of southern France, using several possibilities for the surface representation. In the control experiments, we used the default LC database in WRF, which differed significantly from the actual LC. In addition, sub-grid variability was not taken into account since the model uses, by default, only the surface information from the dominant LC category in each pixel (dominant approach). To improve this surface simplification, we designed three new interconnected numerical experiments with four

widely-used land-surface models (LSMs) in WRF. The first one consisted of using a more realistic and higher-resolution LC dataset over the area of analysis. The second experiment aimed at investigating the effect of using a mosaic approach, where 30-m sub-grid surface information was used to calculate the final grid fluxes, based on weighted averages from values obtained for each LC category. Finally, in the third experiment, we increased the model stomatal conductance for conifer forests, due to the large fluxes errors associated with this vegetation type in some LSMs. The simulations were evaluated with gridded

area-averaged fluxes calculated from five tower measurements obtained during the Boundary Layer Late Afternoon and Sunset Turbulence (BLLAST) field campaign. The results from the experiments differed depending on the LSM and displayed a high dependency of the simulated fluxes on the specific LC definition within the grid cell, an effect that was enhanced with the dominant approach. The simulation of the fluxes improved using the more realistic LC dataset except for the LSMs that included extreme surface parameters for the coniferous forest. The mosaic approach produced fluxes more similar to reality

and served to improve, especially, the latent heat flux simulation of each grid cell. Therefore, our findings stress the need to include an accurate surface representation in the model, including soil and vegetation sub-grid information with updated





surface parameters for some vegetation types, as well as seasonal and man-made changes. This will improve the modelled heat fluxes and ultimately yield more realistic atmospheric processes in the model.





## 1 Introduction

The Earth's surface is constantly changing at different timescales (Sellers et al., 1995). Natural changes of the land surface (vegetation) occur due to climate variability and seasonality (Weltzin and McPherson, 1997; Crucifix et al., 2005). However, human beings have significantly contributed to non-natural and accelerated changes in land cover (LC), especially during recent decades (Pielke et al., 2011). These changes can be extremely important because they modify the natural cycles of

energy (Seneviratne et al., 2010), trace gases (e.g., Muñoz-Rojas et al., 2015; Green et al., 2019) or nutrients (e.g., Holmes et al., 2005), among others, and because they alter ecosystems (Pielke et al., 1998). The consequences of these alterations are difficult to predict due to the non-linearity of the numerous connected processes (Pielke et al., 1999). On the one hand, the changes in the habitat alter food chains and impact vegetal and animal species (Auffret et al., 2018), but also smaller organisms living within them or in equilibrated soil, such as bacteria and viruses (Jeffery and Van der Putten, 2011; Blackburn et al.,

2007; Rulli et al., 2020). On the other hand, radiative and texture properties of the surface are also modified: changes in albedo (Loarie et al., 2011), emissivity, thermal properties of the soil (Luyssaert et al., 2014) or surface roughness (Bonan et al., 2018). This modifies the heat and water exchange processes between the surface and the atmosphere by altering the net radiation at the surface. Indeed, the water transfers from the soil to the air (and vice versa) are significantly linked to the vegetation type and the soil properties: infiltration, runoff, soil moisture or evapotranspiration (ET) (Zhang and Schilling, 2006). All these changes in

the surface energy balance have direct or indirect feedbacks on the planetary boundary layer (PBL) development (Combe et al., 2015), cloud formation (Vilà-Guerau De Arellano et al., 2012), atmospheric temperatures (Koster et al., 2006; Christidis et al., 2013), and rainfall (Koster et al., 2003). This may impact surface characteristics and vegetation activity again and, in the long term, it will restart the whole cycle by changing species (vegetation included), which need adapt to the modified environmental conditions (Pielke et al., 1998), with the direct or indirect associated impacts on the first triggers, the humans (Meyer et al.,

1994; Rulli et al., 2020).

Since LC-change decisions are typically made by local/regional governments (e.g., Sánchez-Cuervo et al., 2012), it is of utmost importance that these organizations understand the direct and indirect implications of various anthropogenic earth-surface modifications. Hence, it is crucial to quantify the uncertainty associated with land-surface representation in weather and climate models, which is the main objective of this paper.

Current weather and climate models rely on parametrisations to represent energy, water and momentum exchanges between the surface and the atmosphere. This is done by coupling land surface models (LSMs) with the atmospheric component of the predicting system. During the last decades, a significant effort has been made to improve LSMs (Cuxart and Boone, 2020). On the one hand, their complexity has been increased with equations that are able to represent the myriad of processes involved in these exchanges (Lawrence et al., 2019). On the other hand, these equations need accurate parameters that describe the

properties of the soil and the vegetation (Cuntz et al., 2016). Both types of improvements need observational measurements from experimental sites and field campaigns to learn about the surface properties and physical processes, as well as to evaluate models.





In this context, numerous scientific initiatives have been conducted to improve the knowledge of the land-atmosphere interaction processes. This has been done through the design of experimental field campaigns: e.g., The Boreal Ecosystem Atmosphere Study (BOREAS, Sellers et al. (1995)), The Global Energy and Water Cycle Experiment (GEWEX, Chahine (1992)), and The Lindenberg Inhomogeneous Terrain - Fluxes between Atmosphere and Surface: a Long-term Study (LITFASS, Beyrich et al. (2002)), among many others. Other initiatives were focused on the inter-comparison of LSMs: e.g., The Project for Intercomparison of Land-surface Parametrization Schemes (PILPS, Henderson-Sellers et al. (1996)) or the The global land-atmosphere coupling experiment (GLACE, Koster et al. (2006)). Recently, the CloudRoots field experiment (Vilà-Guerau de Arellano et al., 2020) offered an integrated multi-scale approach from leaf to landscape measurements complemented with models.

Some specific works have also focused on the effect of LC through the investigation of the impacts of improving the accuracy and resolution of LC database used in the models (e.g., Pineda et al., 2004; Cheng et al., 2013; Santos-Alamillos et al., 2015; Schicker et al., 2016; Jiménez-Esteve et al., 2018). Others have focused on modelling the changes that might occur under the assumption of possible future changes of the surface (e.g., Li et al., 2018; De Meij et al., 2019). These studies stated the importance of having an accurate surface representation in the models to obtain improved simulations of different variables.

In this sense, the present work was firstly motivated by the inaccurate representation of the LC provided by the default LC dataset (International Geosphere-Biosphere Programme from the Moderate Resolution Imaging Spectroradiometer, IGBP-MODIS) in the Weather Research and Forecasting (WRF) model over the area of analysis (southern France), which differed significantly from the LC observed in the area. We hypothesised that this will lead to errors in the simulated surface energy fluxes (specifically, sensible and latent heat fluxes). Besides, the default configuration in WRF only uses the information from the tabulated surface parameters of the LC category with higher percentage of coverage within each grid cell (dominant approach). This may be appropriate for areas with large-enough homogeneous surfaces, but not for the area of study, where the LC has significant heterogeneous patches that might impact the surface fluxes. This influence of the surface heterogeneous patches on the lower troposphere is known as static heterogeneity (e.g., Patton et al., 2005; van Heerwaarden and Guerau de Arellano, 2008) and has also impacts on the PBL processes (e.g., Margairaz et al., 2020a, b), creating new (dynamical) inhomogeneities such as clouds or modified turbulence. This dynamical heterogeneity will also impact the surface, in an interaction known as dynamical heterogeneity (e.g., Lohou and Patton, 2014; Horn et al., 2015).

This study focused on the static heterogeneity impacts on surface fluxes, through the quantification of the changes associated with several improvements made on the representation of the surface in the WRF model, which is the main objective of this work. In order to strengthen the analysis, four widely used LSMs available in WRF were used: 1) Noah (Chen and Dudhia, 2001); 2) Noah-MP (multi physics) (Niu et al., 2011); 3) CLM4 (The Community Land Model v.4) (Oleson et al., 2010), and; 4) RUC (The Rapid Update Cycle) (Smirnova et al., 2016). The different experiments were designed as follows. First, we improved the LC in the area evaluated using the more realistic and higher resolution LC dataset from the *Centre d'Etudes Spatiales de la Biosphère* research laboratory (CESBIO, Inglada et al. (2017)). The results showed a high dependence of the flux on the specific LC categories, which motivated a second experiment including the sub-grid information of the surface, the so-called mosaic approach (e.g., Li et al., 2013). Finally, an additional experiment was carried out due to the extreme biases found in the first two experiments over those pixels mostly covered by conifer trees for some LSMs (Noah-MP and CLM4),





with the aim of diminishing the biases by modifying some parameters associated with the transpiration processes of this LC category.

For the evaluation of the simulations, we took advantage of the large number of instruments deployed during the Boundary-Layer Late Afternoon and Sunset Turbulence (BLLAST) field campaign, carried out in 2011 in southern France (Lothon et al., 2014). The spatial density of eddy-covariance (EC) towers over different vegetation types facilitated the calculation of gridded area-averaged fluxes (AAF), as done in a similar way for the LITFASS experiment (Beyrich et al., 2006), where they obtained a good agreement with the fluxes measured from scintillometry. In the present work, the AAF were used to evaluate the results from the WRF model coupled with the four LSMs under the different conditions set in the experiments, and to analyse the results based on the different LC types.

The article is organised as follows: Section 2 provides information of the measurements taken during the BLLAST field campaign and explains how the area-averaged fluxes were calculated. Detailed information about the model configuration and the different experiments are also included in this section. In Section 3, we quantified the results from the different modelling experiments, including scientific discussion about them. Finally, a short summary and the main conclusions are provided in Section 4.





## 2 Evaluation data, WRF model and experiments design

### 2.1 Observational data for model evaluation

The surface turbulent heat fluxes simulated by the WRF model were evaluated during a period of the BLLAST field campaign
(Lothon et al., 2014). This campaign took place from 14 June to 8 July 2011 on the Plateau of Lannemezan (southern France).
Its main objective was to better understand the turbulence decay observed during the afternoon transition, and the extensive
instrumentation deployment included several surface-energy-balance towers installed over different surfaces, which were representative of the vegetation within the explored area: prairies, forests, wheat, corn and moor. The analysed period comprised
from 09:00 UTC to 15:00 UTC 19 June 2011, corresponding to part of IOP (intensive observation period) 2 of the campaign.
This IOP was characterised by fair weather, no clouds, and a typical development of the boundary layer up to 800-1000 m
above ground level (agl). The IOP is representative of the general conditions of the rest of the IOPs of the campaign (the general meteorological conditions of the campaign can be found in Lothon et al. (2014)). The sensible and latent heat fluxes (SH
and Le respectively) simulated by WRF were evaluated hourly and, in some cases, subsequently averaged. We focused on the
model bias and root-mean-square error (RMSE), using as benchmark data the gridded AAF (area-averaged fluxes) calculated
using the fluxes values observed over different vegetation types.

### 2.1.1 Observed fluxes over different vegetation types

SH and Le were calculated uniformly, using the eddy-covariance (EC) method with the specifications indicated in De Coster
et al. (2011) over five different LC types: grass, wheat, corn, moor and forest (conifers). The position above ground level (agl)
of the instruments was set according to the vegetation height, which implied an homogeneous footprint for the five towers: 2
m for the moor, grass and wheat sites, 4 m for the corn site and 31 m for the forest site (approx. 6 m above the trees). These
measurements showed that the SH was more sensitive to LC type (Fig. 1a and c) than Le (Fig. 1b and d). This was probably
linked to the different surface properties of the vegetation types (albedo, thermal inertia, emissivity), with more influence on the
surface temperatures. The highest SH was measured over the forest site, followed by the wheat site, with peaks of around 400
and 270 $\mathrm{Wm^{-2}}$ respectively. The measurements over the grass, moor and corn sites showed lower values with maximum SH
of around 150 $\mathrm{Wm^{-2}}$ during the central hours of the day. However, Le exhibited values that were similar for all the vegetation
types, with midday maxima around 300 $\mathrm{Wm^{-2}}$. Nevertheless, the lowest Le was measured over the wheat field (this crop started
to dry during the experiment), while the largest values were observed at the grass, forest and moor sites. Some differences were
observed in the diurnal evolution of Le: the morning Le rise over the corn/wheat were delayed with respect to the measurements
over other LCs, maybe linked to a delay in the transpiration processes associated with these crops (a full investigation of the
reasons of it is out of the scope of this study). In any case, it should be highlighted that these fluxes values were representative
of those of the different IOPs of the BLLAST field campaign (Lothon et al., 2014; Couvreux et al., 2016), with no-water
limitation due to the regular rain events observed in the area during the weeks before, which are typical conditions in these
dates in the area.



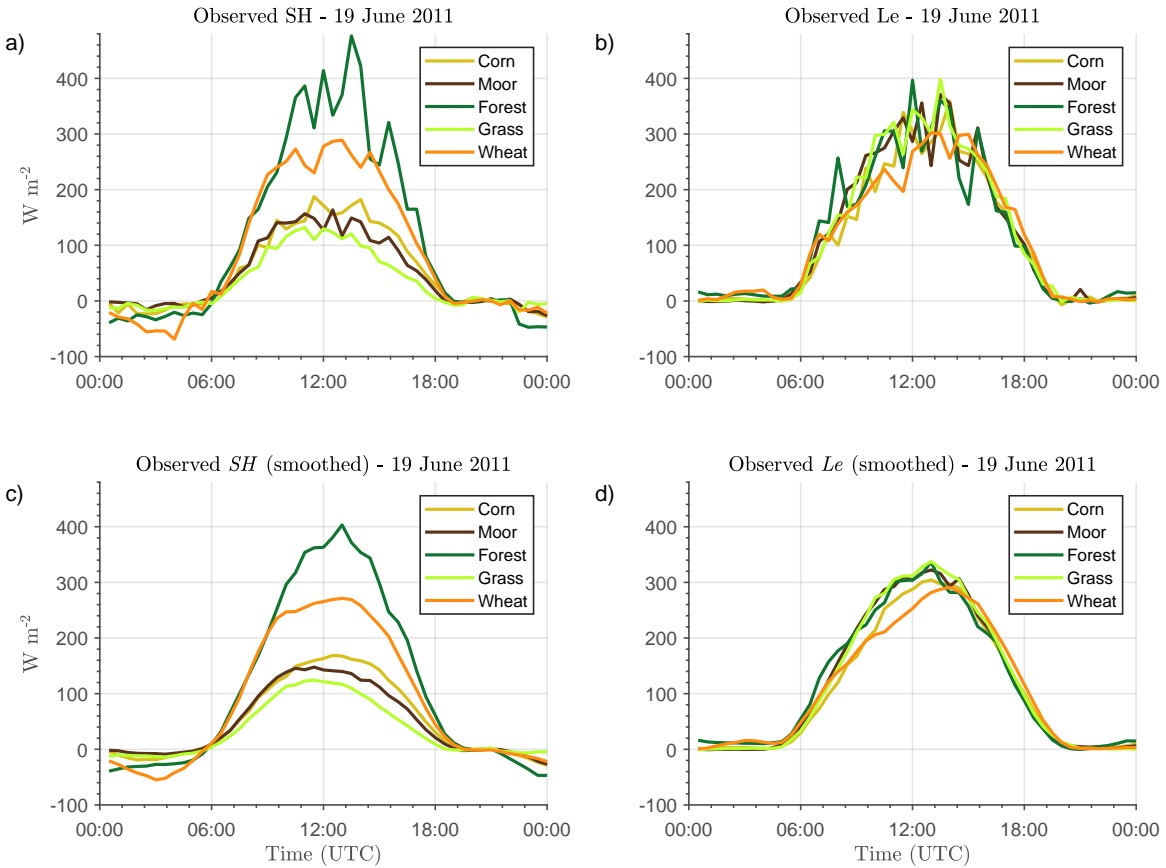

**Figure 1.** a) Observed sensible heat flux (SH) on 19 June 2011 during the BLLAST campaign over different vegetation types. b) Idem for the latent heat flux (Le). c, d) Same as in a and b, but with the time series smoothed for a better visualization of the main differences between vegetation types. The fluxes values were calculated using the eddy covariance technique with averaging windows of 30 minutes.

### 2.1.2 Area-averaged fluxes (AAF) calculation

The EC measurements from towers were used for the calculation of AAF with resolution of 1 km in a region of 19x19 km around the central site of the field campaign: $43°07'27.15''N, 0°21'45.33''E$; 600 m above sea level (asl). This approach can be considered as a mosaic observational method for flux computation. It was based on the multiplication of the values of the fluxes measured over each vegetation type by their respective vegetation-cover fraction within each 1x1 km pixel (see Fig. 2a, b). Finally, the contribution from each vegetation type was summed to provide the final values in each grid cell (area-averaged

fluxes). High-resolution and realistic LC data were needed to construct the AAF, obtained from the 30-m LC database prepared by CESBIO, based on Landsat-5 data (Inglada et al., 2017) (Fig. 2b). Hence, the main assumption of the AAF calculation was to





consider the same SH and Le for sites with the same LC type at different locations, implying also the following simplifications (further discussion is included after the list):

1. There were no flux differences depending on the soil type.

2. There was no difference in activity for the same vegetation type in the area.

3. There was no horizontal variability of the soil moisture (SM).

4. The radiation and the wind forcing was the same for all the pixels, without altitude influences.

5. All the forests were considered coniferous (evergreen needle-leaf forest (ENF)).

6. Fluxes over urban and bare-soil surfaces were estimated using a simple model.

(1) Two soil types were the dominant ones in the evaluation area, based on the 1x1 km pixels from the soil database of WRF (United States Department of Agriculture, USDA): clay loam (81% of the area) and loam (19% of the area). Since the dominant soil type in the pixels where the measurements were taken was clay loam, those pixels with loam were not included in the evaluation of the model. Therefore, the limitation based on the soil type was ultimately not a problem. This area is shown in Fig. 2d with a black rectangle.

(2) The area analysed is relatively small to expect significant differences between the natural vegetation belonging to the same category. Although the limitation based on the possible different vegetation state can be more important in crop areas, seeding dates are normally coincident between fields in the area.

(3) Regarding the effect of possible SM differences within the area, two aspects should be noted. On the one hand, the potential SM horizontal variability due to possible inhomogeneous precedent rainfall over the area was not taken into account. However, the SM input in the model is provided with a coarse resolution of 1° and does not show any small-scale details. This is a well-known limitation of mesoscale models which is sometimes addressed through the assimilation of SM data from satellites or with previous long simulations that serve to spin-up the surface in order to obtain the appropriate SM initial values (De Rosnay et al., 2013; Angevine et al., 2014; Santanello et al., 2016). In our case, this limitation of the mesoscale modelling is an *advantage* because it allows to perform a fairer model-observation comparison since this limitation also exist in the AAF.

On the other hand, part of the SM horizontal variability is caused by the different properties of the vegetation patches, affecting runoff, infiltration, evapotranspiration, etc., and, finally, the SM content associated with each vegetation type. This LC effect on SM may be implicitly included in the fluxes that were measured over each individual vegetation type. In any case, we investigated the effect of including the SM horizontal heterogeneity with a higher resolution in the model initialization (not shown). This was done by using high-resolution SM satellite data from the Disaggregation based on Physical And Theoretical scale Change (DISPATCH) (Merlin et al., 2013; Molero et al., 2016) product at 1 km of resolution, but conserving the original range of SM variability within the area evaluated. The effect of including this SM spatial heterogeneity was minimum in the fluxes simulations (in part due to the conservation of the range of SM values). Even if dedicated studies could be designed to





investigate the effect of initialising the model varying the range of SM values in the area, these studies deserve a fully dedicated effort and are out of the scope of this paper.

(4) The limitation due to the radiation and wind forcing, which were assumed as equal in all the pixels, is expected to have a small impact, due to the fair weather, with no clouds, and light wind conditions on the 19 of June (Lothon et al., 2014). However, some differences could exist between some pixels situated at different altitude over the area (most of the pixels were in the range of altitude between 400 and 650 m asl, with minimum and maximum altitude of 282 m and 696 m asl in the whole evaluated area).

(5) The limitation of considering all forests as coniferous was due to the fact that the campaign only included measurements over this type of trees, and not over broadleaf deciduous forests (the other forest category in the area). Therefore, the measurements taken over conifers were extrapolated to those areas covered by deciduous for the calculation of the AAF. This limitation was addressed by converting all the forest in the WRF model to conifer, which allowed a fairer model-observations comparison at the expenses of loosing the analysis over one LC category (deciduous forest).

(6) For the urban surfaces, SH and Le were calculated following a simple Penman–Monteith type model (De Bruin and Holtslag, 1982) shown in Eq. (1) and Eq. (2), respectively:

$$SH = \beta \cdot \frac{R_n - G}{1 + \beta}, \tag{1}$$

$$Le = \frac{R_n - G}{1 + \beta}, \tag{2}$$

where $R_n$ (Eq. 3) is the net radiation, defined as:

$$R_n = (1 - \alpha) \cdot SW \downarrow + LW \downarrow - \epsilon \, \sigma \, T_s^4, \tag{3}$$

and $G$ (Eq. 4) is the ground heat flux, considered a fixed fraction of $R_n$:

$$G = G_{frac} \cdot R_n. \tag{4}$$

In these equations, $SW \downarrow$ and $LW \downarrow$ are the measured downward short-wave and long-wave radiative fluxes, respectively. The albedo ($\alpha$), emissivity ($\epsilon$), the fraction of energy used for ground heat flux ($G_{frac}$), and the Bowen ratios ($\beta$) were 200 considered constant, with values of 0.15, 0.92, 0.3 (stable)/0.5(unstable) and 5, respectively, obtained from Lemonsu et al. (2004) and Grimmond and Oke (1999) for urban surfaces. These simplifications could have led to some overestimation of urban effect on the total fluxes, since the urban surfaces in the area were surrounded by gardens, prairies and trees (so-called urban diffuse in the CESBIO LC dataset). However, the 30-m resolution LC dataset should be appropriate to deal with the urban/vegetation patches.





Despite the high-quality data and the efforts carried out to reduce uncertainties in the AAF, the evaluation of models with surface observations is a necessary task which is not straightforward. The observational measurements are almost always linked to uncertainties that can add limitations to the evaluation process (Bou-Zeid et al., 2020). In our case, the measurements were taken at heights that implied homogeneous footprints, but even in this case, other uncertainties are always present (Mauder et al., 2020), especially in the case of the measurements taken over vegetation with tall canopies, where the heat storage can

have an important role in the surface energy balance.

### 2.1.3 Additional data for evaluation of the PBL height

In order to follow-on studies to the present work, a preliminary analysis of the impacts of the surface changes on PBL height ($z_i$) is included in Section 4. To validate the simulation of this variable, different estimated values have been used from BLLAST observations, which are specified and explained in Table 1. Most of these measurements were taken at the central site of the

campaign (Site 1), but the measurements from a radiosounding (RS2) launched at an additional site (Site 2), located 4 km to the south of Site 1, were also used.

**Table 1.** Summary of the different estimations of the PBL height ($z_i$) from BLLAST observations. The values specified are those used to be compared with the $z_i$ simulated by the WRF model at 12:00 UTC. * Small Unmanned Meteorological Observer (SUMO). ** Ultra-high frequency wind profiler.

| $z_i$ name | Instrument | Site | Time | $z_i$ | Technique |
|---|---|---|---|---|---|
| RS1 | Modem radiosounding | Site 1 | 11:04 UTC | 836 m | Virtual potential temperature gradient |
| SUMO* | Unmanned aerial vehicle | Site 1 | 11:12 UTC | 823 m | Virtual potential temperature gradient |
| UHF | UHF** Degreane-Horizon | Site 1 | 12:04 UTC | 1100 m | Local maxima of refractive index structure coefficient |
| LIDAR | LIDAR | Site 1 | 12:06 UTC | 1006 m | Aerosol, filtered wavelet covariance transform method |
| RS2 | Vaisala radiosounding | Site 2 | 13:00 UTC | 919 m | Virtual potential temperature gradient |





## 2.2 WRF model

The WRF-ARW (Weather Research and Forecasting-Advanced Research WRF) v.4.1.3 (Skamarock et al., 2019) was used to run the different numerical experiments. The model ran for 60 hours (from 12:00 UTC of day 18 June 2011 to 00:00 UTC of

day 21 June) but only the central hours of day 19 June were evaluated. The first 21 hours of the simulation were used as spin-up and the diurnal cycle of day 20 June was not analysed due to the presence of clouds in the area, which made it more difficult to carry out the strategy designed for the main objective (Pedruzo-Bagazgoitia et al., 2017). The simulations were configured with four nested domains of 27, 9, 3 and 1 km of resolution.

The inner domain covered an area of 120x120 km², but only an area of 19x19 km² around the central point was evaluated,
which had the same grid as the AAF used for the evaluation, i.e., the centre of the central pixel was located at the same location as the centre of the AAF used for the evaluation. More details about the WRF technical configuration common for all experiments are included in Table 2.

**Table 2.** Details about the WRF model configuration common to all simulations. *National Centers for Environmental Protection (NCEP) FNL (final) Operational Model Global Tropospheric Analyses, continuing from July 1999 (NCEP, 2000).

| | |
|---|---|
| Model version | WRF-ARW v.4.1.3 |
| Number of domains | 4 |
| Resolution of domains | 27/9/3/1 km |
| Initial and boundary data | NCEP-FNL* data (1°), each 6 hours |
| PBL scheme | Yonsei University (YSU, Hong et al. (2006)) |
| Surface-layer scheme | MM5 similarity (Jiménez et al., 2012) |
| Land-surface models | Noah / Noah-MP / CLM4 / RUC |
| Microphysics scheme | WRF Single-Moment 3-class (Hong et al., 2004) |
| Long-wave radiation scheme | Rapid radiative transfer model (RRTM, Mlawer et al. (1997)) |
| Short-wave radiation scheme | Dudhia (Dudhia, 1989) |
| Number of vertical levels | 40 |
| Time step | 90/30/10/3.3 s |
| Model initial date | 18 June 2011 at 12:00 UTC |
| Period analysed | 19 June 2011 (09:00 - 15:00 UTC) |
| Leading time (spin-up) | 21 h |

### 2.2.1 WRF land surface models (LSMs)

In order to add robustness to the study, four different LSMs available in WRF were analysed. The information below is mostly
extracted from the literature:





1. Noah (Chen and Dudhia, 2001). Noah is a widely used LSM resulting from the collaboration among many different institutions. It is the default option in WRF and it is used in many other models, with an important application in operational models from the National Centers for Environmental Prediction (NCEP). The model considers four soil layers, where it computes temperature and soil moisture. It takes into account the type of vegetation (LC category), monthly vegetation fraction and soil type to calculate the runoff, ET and root zone. Since the WRF v.3.6, there is a mosaic option available in the model (Li et al., 2013) to deal with the sub-grid heterogeneity (this option is not activated by default).

2. Noah-MP (Noah Multi-physics) (Niu et al., 2011). It is an extension of the Noah LSM that allows the use of multiple options for land-atmosphere processes (e.g., infiltration, run-off, etc.), resulting in a total of more than 4000 combinations (the default options are used in this work). This LSM contains a separate vegetation canopy with a two-stream radiation transfer approach, shading effects and complex physics for the snow/ice processes within the soil. This LSM uses a different set of parameters for each vegetation category than Noah, with more vegetation-dependent parameters.

3. CLM4 (The Community Land Model v.4) (Oleson et al., 2010). It is a sophisticated LSM including state-of-the-art scientific knowledge about soil-vegetation-atmosphere processes. It first divides the simulation into five cover types (glaciers, lake, wetland, urban and vegetation). The vegetation is subsequently divided into four different plant functional types (PFTs) that are defined based on the LC categories. It should be noted that this LSM was computationally the most expensive among the four analysed in this work.

4. RUC (The Rapid Update Cycle) (Smirnova et al., 2016). It uses 9 soil layers with higher density close to the surface. It has a complex treatment of snow processes. In the warm season, it corrects soil moisture in cropland areas to compensate for irrigation. This model also allows a mosaic approach for the sub-grid treatment of the cell heterogeneity, but it is different than in Noah, with albedo values that correspond only to those parameters associated with the dominant LC category. The vegetation parameters used are obtained from the same look-up table than in Noah.



## 2.3 Experiments design

We performed a set of four different modelling experiments aimed at checking the sensitivity of the model to different changes
in the representation of the surface. These experiments are summarised in Table 3 and explained below:

**Table 3.** Summary of the simulations and the names used along the article. Note how some experiments were not possible (-) for some LSMs.

| EXPERIMENTS | Noah | Noah-MP | CLM4 | RUC |
|---|---|---|---|---|
| **DEFAULT**<br>Default setting | DEFAULT-Noah | DEFAULT-Noah-MP | DEFAULT-CLM4 | DEFAULT-RUC |
| **NEW-LC**<br>More realistic and higher resolution LC | NEW-LC-Noah | NEW-LC-Noah-MP | NEW-LC-CLM4 | NEW-LC-RUC |
| **MOSAIC**<br>NEW-LC and mosaic approach | MOSAIC-Noah | - | - | MOSAIC-RUC |
| **FOREST**<br>NEW-LC and conifer transpiration increased | - | FOREST-Noah-MP | - | - |

### 2.3.1 DEFAULT experiment

In this experiment we used the default configuration for the surface representation in WRF, i.e., the LC dataset from IGBP-
MODIS (21 categories) and a dominant approach for the sub-grid heterogeneity, which means that the fluxes were calculated
taking into account only the surface parameters of the dominant LC category, i.e., assuming that no sub-grid variability exists
even when this information is available. The dominant LC category is the one with the highest percentage of coverage within
each pixel. This method has an intrinsic dependency on the number of LC categories of the dataset. For example, a pixel with
40% of water, 30% of conifer forest and 30% of deciduous forest will be treated as water, since it is the dominant category
despite both types of forest cover 60% of the total surface of the grid. On the contrary, the dominant category would be forest
if both types of forest were merged into a single category.

It is expected that these simulations will be limited by the fact that the representation of the LC in the area by IGBP-MODIS
is not totally correct in comparison with the reality (shown later). Besides, the dominant approach implies that the model does
not take advantage of the higher resolution of the LC datasets.

### 2.3.2 NEW-LC experiment

In this experiment, we used a more realistic LC dataset than IGBP-MODIS, obtained from 30-m resolution data prepared by
CESBIO (Inglada et al., 2017). Figure 2a shows a satellite image of the area evaluated, which serves to visually validate the
CESBIO LC at 30-m resolution (Fig. 2b). The inaccurate representation of the pixel-dominant LC by IGBP-MODIS is revealed



in Fig. 2c and the more appropriate surface representation of the dominant LC categories of each pixels used in the NEW-LC experiment is shown in Fig. 2d.

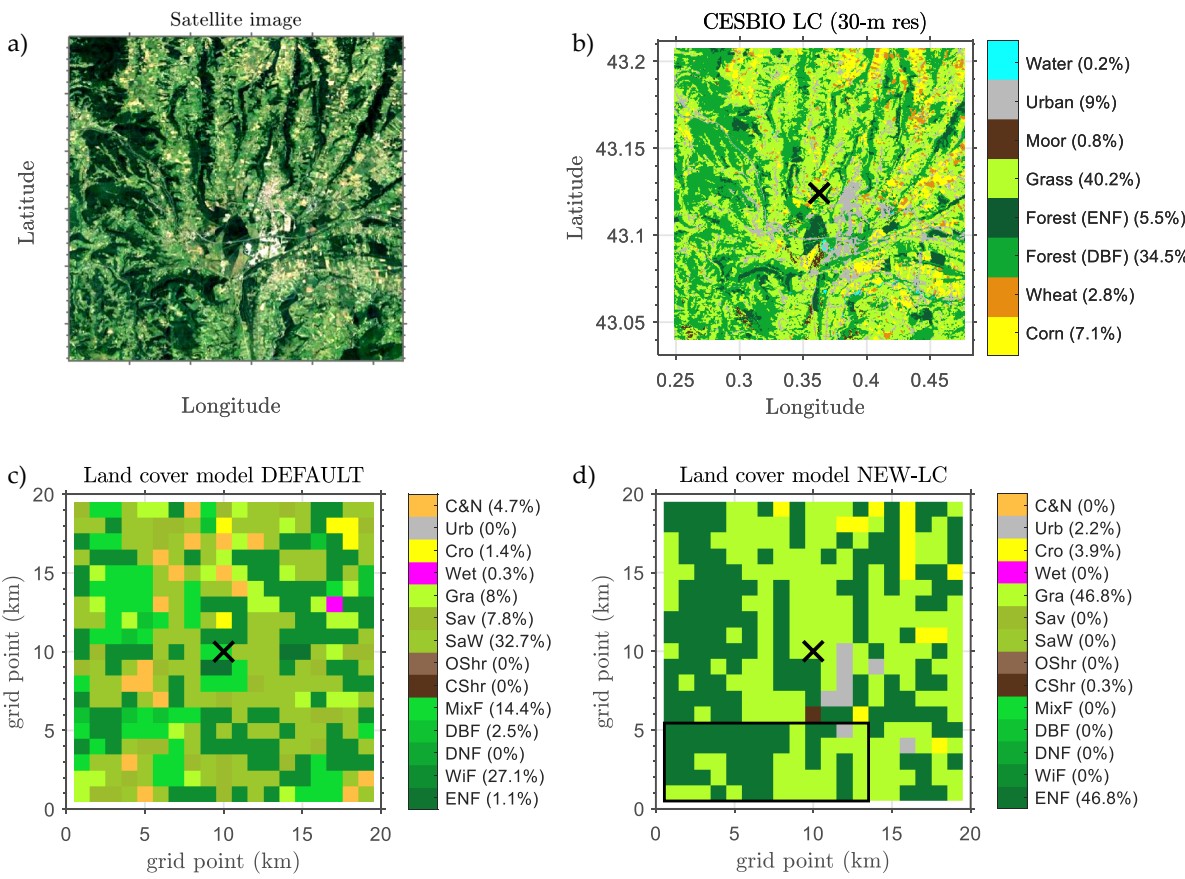

**Figure 2.** a) Satellite picture of the area analysed (From ©Google Earth, image Landsat/Copernicus) b) CESBIO land cover map from 30-m resolution data. c) Dominant land cover (1 km) used in the default experiment (DEFAULT), from the IGBP-MODIS database. d) Dominant land cover (1 km) used in NEW-LC experiment, obtained from the CESBIO land cover map shown in b. The black rectangle indicates the area with soil type dominated by loam, which was not used for the evaluation. The rest of the area was characterised by clay loam (dominantly), as the pixels including the sites where the EC measurements were taken (maximum 4 km away from the central point of the BLLAST field campaign, indicated with a black x symbol). The complete names corresponding to each LC category abbreviation are included in Appendix A.

In order to incorporate the more realistic LC from CESBIO in the WRF model, first, the different categories of the new
LC dataset (CESBIO, 17 categories) were transformed into the most appropriate ones of the WRF default LC dataset (IGBP-MODIS). That is, the same 21 LC categories of IGBP-MODIS were conserved, following a similar approach as done in





Pineda et al. (2004) and in Schicker et al. (2016). We used the transformations indicated in Appendix A, with some special considerations:

- The two types of forest distinguished by CESBIO (conifer and deciduous) were transformed to conifer (evergreen needle-leaf forest, ENF), since the EC measurements used in the AAF were taken over this type of forest due to the lack of measurements over deciduous broadleaf forest (DBF). As commented before, the measurements taken over conifers were extrapolated to the areas with deciduous forests in the AAF calculation. By converting all the forest to conifer also in WRF, the observations-model comparison was fairer.

- The two possible types of crop in CESBIO (winter and summer crop type, i.e., wheat and corn respectively) were transformed to the single cropland category available in IGPB-MODIS. Hence, we did not take advantage of the differentiation of crop type by CESBIO, but it is important to note that wheat surfaces only covered 2.8% of the analysed area (Fig. 2b).

- The four different urban categories of CESBIO were transformed to the single urban definition of IGBP-MODIS. Most of the urban surfaces of the area evaluated were defined as diffuse urban (8.2%) and industrial zones (0.8%).

This new LC information was incorporated in the modelling system following the technical details included in Appendix B. Note how even using a more realistic LC dataset over the area, the sub-grid information was not used because the dominant approach was maintained in this experiment.

### 2.3.3 MOSAIC experiment

The sub-grid heterogeneity is important in the case of the BLLAST area due to the small scale of the LC patches (see Fig. 2b). This makes the area very appropriate for investigating the use of a mosaic approach in the model, as done in the labelled MOSAIC experiment. The mosaic approach implies that the flux is computed for each LC category (tiles) within each grid cell based on the surface parameters tabulated for each one; and then averaged taking into account the percentage of coverage of each tile. That is, taking into account the sub-grid surface heterogeneity. In this context, Noah (see Li et al., 2013, for a complete description of the Noah mosaic approach) and RUC allow the possibility of using this approach in the WRF modelling system. The LC improvements of the NEW-LC experiment were also included here. More technical details about the model implementation of this experiment are included in Appendix B.

### 2.3.4 FOREST experiment

The last experiment was motivated by the large surface fluxes errors found over those pixels covered by conifer (ENF) in previous simulations using Noah-MP and CLM4. Specifically, these LSMs overestimated the SH and underestimated the Le. We hypothesised that this was caused by a parametrized resistance to transpiration that was to high. Hence, we designed the FOREST experiment for Noah-MP. We modified three parameters for ENF, with the values used in Bonan et al. (2014):





1. The slope of the Ball Berry conductance (the inverse of the stomatal resistance to transpiration) equation, the so-called MP in Noah-MP and usually known as $g_1$ (the slope parameter). The Ball-Berry equation (Ball et al., 1987) linearly relates the stomatal conductance to the $CO_2$ assimilation rate:


$$g_s = g_0 + g_1\,A_n \cdot \frac{h_s}{c_s}. \tag{5}$$

In this equation, the slope ($g_1$) represents the sensitivity of the stomatal conductance to assimilation, $CO_2$ concentration, humidity and temperature. It is the parameter that most affects the plant's transpiration (Cuntz et al., 2016), which increases with larger slope values. The $g_0$ parameter is the minimum conductance, $h_s$ and $c_s$ are the fractional relative humidity and the $CO_2$ concentration at the leaf surface, respectively, and $A_n$ is the net leaf $CO_2$ assimilation.

MP has a default tabulated value of 6 for ENF for Noah-MP and for CLM4 (Oleson et al., 2010), a value that is significantly different compared to the values assigned for other categories (9 for most of them, including, for example, the broadleaf evergreen forest (BDF)). The lower value for ENF limits the transpiration processes and could be the reason for part of the Le underestimation, leading to more energy being distributed to SH. This parameter was optimised in Bonan et al. (2014) and a value of 9 was also used for ENF in their study.

2. The minimum leaf conductance or the interception in the Ball-Berry equation ($g_0$, indicated as BP in the Noah-MP scheme). This parameter was changed from 0.002 to 0.01 $mol\,H_2O\,m^{-2}s^{-1}$, as in Bonan et al. (2014) for ENF.

3. The maximum carboxylation rate at 25 ºC ($V_{cmax25}$), a photosynthetic parameter that was changed from 50 to 62.5 $\mu mol\,m^{-2}s^{-1}$, as used in Bonan et al. (2014), which value was also selected according to previous literature. The range of observational values in their work for ENF ranged from 48 to 72 $\mu mol\,m^{-2}s^{-1}$ and a value of 62.5 $\mu mol\,m^{-2}s^{-1}$ was

finally selected.

Hence, these three parameters were modified in Noah-MP according to Bonan et al. (2014), following the technical details included in Appendix B. Indeed, these changes allowed for more evapotranspiration in the ENF forest, which should improve the evaluation in our case study. It should be noted that the $g_1$ parameter (MP) was the one which most influenced the results, as indicated in Cuntz et al. (2016) and as observed in additional previous experiments performed in our case (not shown). In

any case, the objective of this experiment was to demonstrate the high impact of the associated vegetation parameters on the surface fluxes, not the optimization of these values for the specific tree species present in the area here analysed.





## 3 Results: Turbulent fluxes sensitivity to surface changes

The model skill simulating the AAF fluxes is quantified in this section for the different experiments listed in Table 3. The results are shown and commented individually for each experiment. Since some of the results and the associated discussion differed significantly depending on the LSM used, the results were subdivided for each LSM. As a general reference, Table 4 shows a summary of the total RMSE obtained for SH and Le using the different LSMs and for all the experiments performed.

**Table 4.** Summary of the root-mean-square error (RMSE) in $Wm^{-2}$ calculated for the evaluated area for each LSM (columns) configured for each experiment (raws). The first value corresponds to the sensible heat flux (SH) RMSE, and the second one to the latent heat flux (Le) RMSE. Some experiments were not possible (-) for some LSMs.

| Experiment | RMSE ($Wm^{-2}$) − SH/Le | | | |
|---|---|---|---|---|
|  | **Noah** | **Noah-MP** | **CLM4** | **RUC** |
| **DEFAULT** | 56 / 76 | 96 / 46 | 69 / 55 | 97 / 81 |
| **NEW-LC** | 44 / 63 | 116 / 90 | 91 / 82 | 90 / 85 |
| **MOSAIC** | 43 / 45 | - | - | 85 / 76 |
| **FOREST** | - | 77 / 56 | - | - |

### 3.1 DEFAULT experiment versus NEW-LC experiment

#### 3.1.1 Noah

The bias of the simulated SH and Le by the DEFAULT-Noah simulation relative to the observations (the AAF) is shown in Fig. 3a and d respectively. The results are presented for each LC category, taking into account the dominant LC of each grid cell (1 km) in the model. Note that the DEFAULT experiment includes ten LC categories in the area, based on the IGBP-MODIS database. The total bias values for SH and Le (represented with blue dashed horizontal lines) are close to 0 $Wm^{-2}$ due to the compensation effect when merging cells with positive and negative biases. In any case, the near-zero value indicates that the averaged values over the entire area are close to those that would be obtained from the observations. However, there are some LC categories that systematically present noticeable biases. Those pixels represented as WiF (Wild forest) in the model (27% of the total area) normally present an overestimation of the SH, while the Mixed forest (MixF) pixels (14%) underestimate it (Fig. 3a). For the Le, the pixels covered by SaW (Savanna Woody, which represent 33% of the area) are normally characterised by a remarkable Le underestimation, with biases values around -100 $Wm^{-2}$ (Fig. 3d).

The definition of some of the LC categories existing in the DEFAULT experiment did not represent appropriately the real LC of the area (as seen in Fig. 1c), but they influenced notably the simulated model errors. This is the first indicator of the high





dependency of the model results on the LC categories. Then, based on the main hypothesis, the results should improve using a more realistic LC representation (NEW-LC experiment).

The general results for SH and Le from NEW-LC-Noah are shown in Table 4. In this experiment, the LC was modified towards a more realistic representation using the high-resolution data from CESBIO. This led to five LC categories that were
present in the area of study as dominant LC, with evergreen needle-leaf forest (ENF) and grass (Gra) covering the 45% and 47% of the total area, respectively. The SH biases observed in the DEFAULT experiment were improved in the NEW-LC, especially for ENF and Gra, with slightly negative (close to zero) values. This led to a slightly negative SH bias in the whole area (Fig. 3b, see horizontal red dashed line). For the Le bias (Fig. 3e), the opposite is observed, with a slight Le overestimation mainly caused by the Le overestimation over ENF (too much ET simulated by the model in this forest type). As it will be shown later,
this result contrasts with the findings obtained with the rest of LSMs.

The bias is a good indicator of systematic errors for specific LC categories, but it is a poor indicator of the total model behaviour in the area (it can mix positive and negative values leading to neutral ones). Hence, the RMSE has been included in Fig. 3c and f, as an indicator of the total performance of the NEW-LC experiment. Note how in the case of the RMSE (Fig. 3c and f), the results are shown by real LC categories, whose fractions of coverage are very similar to those used in the NEW-LC
experiment (slightly different due to small differences in the categories, e.g., corn and wheat in the real LC were merged into crop in the model).

The SH-RMSE (Fig. 3c) improved for the total area when using the improved LC dataset, from 56 $\mathrm{Wm^{-2}}$ (DEFAULT-Noah) to 44 $\mathrm{Wm^{-2}}$ (NEW-LC-Noah). The vertical bars of Fig. 3c indicate the RMSE for each real LC category (the LC categories used in the AAF). They provide information about the types of vegetation associated with larger errors, i.e., vegetation types whose
processes or parameters are not well represented by the model. In this case, the SH improvements in the NEW-LC experiment (red bars) are observed for all the pixels except for the only pixel where the dominant LC is wheat, associated with a larger SH flux than corn in the observations (see Fig. 2). In any case, wheat crops only covered 0.3% of the total area as dominant LC category (one pixel) and, therefore, the contribution to the total RMSE values was small. The results for Le-RMSE (Fig. 3f) also show significant improvements when using the improved LC dataset, except for those pixels covered by urban land use,
where the Le was notably underestimated (see Urb in Fig. 3e), but with a small impact in the total values due to their scarce presence (2.4%) as dominant pixels. The Le improvement was more substantial in those pixels covered by grass than by forest, since Le is slightly overestimated in the forest pixels (see ENF in Fig. 3e). Notice how besides the better LC pixels distribution in the NEW-LC experiment, we also avoided using some LC categories associated with large errors in the DEFAULT-Noah experiment, such as the unrealistic SaW pixels related to large Le underestimation (Fig. 3d).





# Noah

**Figure 3.** a) Sensible heat fluxes (SH) bias boxplots by model LC category (x-axis) for the default Noah simulation (DEFAULT-Noah). Values are calculated for all the pixels with the specific LC category as dominant in the entire analysed area from 09:00 UTC to 15:00 UTC (hourly values). The horizontal black solid line indicates the line of 0 $\mathrm{Wm^{-2}}$ bias. Horizontal dashed line indicates the mean bias in the area analysed. The boxplots indicate the 50% of the distribution (from percentile 25 to percentile 75), the upper and lower whiskers the rest of the distribution and the red crosses the outliers. Note that these figures include the variability due to the number of pixels with the same LC category as dominant (variable depending on their coverage) and also the variability due to the seven hours analysed. b) Idem for the results obtained from the Noah simulation with the improved LC (NEW-LC-Noah). c) SH root-mean-square error (RMSE) for DEFAULT-Noah and NEW-LC-Noah for each real LC in the area. Horizontal dashed lines indicate the mean RMSE for the whole analysed area. d, e, f) Idem than a, b, c but for the latent heat flux (Le). The percentages included in the x-labels refer to those covered by each category with respect to the total evaluated area.





### 380  3.1.2  Noah-MP

Most of the LC categories showed a positive SH bias for DEFAULT-Noah-MP (Fig. 4a) and a slight negative Le bias (Fig. 4d), especially for those pixels which dominant LC was some of the different types of forest (ENF, WiF, DBF, MixF or SaW). This led to a large SH error and to the smallest error amongst the LSMs compared for Le (despite the slight underestimation), with RMSE values of 96 $\mathrm{Wm^{-2}}$ and 46 $\mathrm{Wm^{-2}}$, respectively (see Fig. 4c and f and also Table 4 for the RMSE comparison among the
four LSMs used).

Contrary to what happened with Noah, the results were, in general (all domain), worse for the NEW-LC experiment, even for the case of SH where the RMSE was already high. The SH bias remained positive, mainly influenced by the significantly high SH bias in ENF pixels (more than +150 $\mathrm{Wm^{-2}}$), while the bias over the grass and crop pixels were close to zero (Figure 4b). For the Le, important negative biases were observed in those pixels mostly covered by ENF (around -120 $\mathrm{Wm^{-2}}$), but with
lower errors for grass or crop pixels. The urban pixels were characterised by important positive SH biases and negative Le biases, leading to RMSE of more than 150 $\mathrm{Wm^{-2}}$ (see red bar for urban in Fig. 4 c and f) for both flux. In any case, the small proportion of pixels with urban as dominant LC category led to a small contribution to the total error.

In contrast, since 45% of the total area was covered by ENF, the final RMSE values of NEW-LC-Noah-MP (horizontal red dashed line in Fig. 4c) are significantly higher than those of DEFAULT-Noah-MP (shown in blue), increasing the total
SH-RMSE up to 116 $\mathrm{Wm^{-2}}$ and the Le-RMSE up to 90 $\mathrm{W\,m^{-2}}$, with a significant worsening for the forest and urban pixels. However, the bars in Fig. 4c and f also illustrate the improvement in the crop, moor and grass pixels for SH and grass for Le, but they were not enough to compensate for the large errors over ENF pixels. This was not observed in the Noah experiment (previous subsection), which uses a different set of vegetation parameters than Noah-MP for each LC category. This suggests that the biased results for Noah-MP could be partially caused by some vegetation parameters that were inappropriate for the
ENF LC category, hypothesis investigated latter in the FOREST experiment.





# Noah-MP

**Figure 4.** Same as in Fig. 3 but for Noah-MP





### 3.1.3 CLM4

The total area-averaged biases for SH and Le were close to 0 Wm$^{-2}$ for the DEFAULT-CLM4 experiment (Fig. 5a and d). The pixels covered by forest in the model tended to overestimate the SH, while those covered by grass, savanna and shrub slightly underestimated it (Fig. 5a). For the Le, there is a general tendency towards slightly underestimated ET processes in CLM4

in comparison with the real observations, except for the WiF (wild forest) and the Wet (wetlands) pixels (Fig. 5d) that were present in the default IGBP-MODIS database. The total RMSE by DEFAULT-CLM4 was 69 Wm$^{-2}$ and 55 Wm$^{-2}$ for SH and Le, respectively (see horizontal dashed blue lines in Fig. 5 c and f, and Table 4).

The NEW-LC-CLM4 simulation (Fig. 5b and e) produced larger biases than the default one, mainly caused by the SH overestimation and the Le underestimation in the pixels covered by ENF (evergreen needleleaf forest), around +100 Wm$^{-2}$ and -

100 Wm$^{-2}$, respectively. The biases observed over grass and crop pixels were close to 0 Wm$^{-2}$ while some issues were observed in those pixels mostly covered by urban, which showed, surprisingly, a substantial underestimation in SH and overestimation in Le of the order of 200-250 Wm$^{-2}$, but with a small effect in the total values due to the low number of pixels with urban as dominant LC.

The large SH overestimation and Le underestimation over those pixels covered by forest (ENF) affected the total performance

of the model, worsening the SH results up to a SH-RMSE of 91 Wm$^{-2}$ and 82 Wm$^{-2}$ for the Le-RMSE. However, as observed in the bars of Fig. 5c and f, most of the LC categories showed SH and Le improvements or almost no-changes in the NEW-LC experiment, except the forest and the urban pixels, whose tabulated parameters seem to be quite extreme, leading to significant errors in the simulation of the fluxes.

These results coincided with those obtained with Noah-MP for the case of ENF grid cells. Indeed, both LSMs employ the

Ball-Berry equation to deal with the stomatal resistance of the plant, and both LSMs use a tabulated value of 6 for the slope of this equation (MP or $g_1$, see Section 2.3.4) for the ENF LC category, while it is 9 for almost all of the rest of the LC categories. Hence, these two LSMs significantly overestimate SH and underestimate Le over the conifer trees in this case study.



# CLM4



**Figure 5.** Same as in Fig. 3 but for CLM4





### 3.1.4 RUC

The SH and Le biases obtained when using the DEFAULT-RUC simulation are the highest ones among the compared LSMs

(see also RMSE in comparative Table 4), with SH-RMSE of 97 $Wm^{-2}$ and Le-RMSE of 81 $Wm^{-2}$. There is a systematic SH overestimation for all the LC categories (Fig. 6a) for DEFAULT-RUC, which is especially aggravated by those pixels unrealistically represented by WiF (Wild forest), with a bias of more than 100 $Wm^{-2}$ which covered an important fraction of the total analysed area (27%). The Le is systematically underestimated (Fig. 6d) for all the LC categories, which shows a tendency towards too little ET in this LSM in this studied case, especially in those LC categories with shorter vegetation (SaW,

Sav and Gra).

These biases were not corrected in the NEW-LC-RUC experiment with the more realistic LC (SH-RMSE of 90 $Wm^{-2}$ and Le-RMSE of 85 $Wm^{-2}$). The values obtained for each LC category (Fig. 6b and e) were within the same order of magnitude than those obtained in the DEFAULT-RUC experiment, or even higher, as it is the case of the SH simulated in the grass surfaces, with a remarkable overestimation of around +75 $Wm^{-2}$. This led to very similar (and high) RMSE values for the DEFAULT

and NEW-LC experiments (Fig. 6c and f). Although the SH simulation improved for the grass and corn pixels, it worsened for the forest pixels. For the Le, the contrary was observed: the biases of the forest and crop pixels improved but worsened for the grass ones, which were associated with too low ET. Note how the RMSE in the NEW-LC experiment over the grass surfaces was higher than 100 $Wm^{-2}$ in RUC while they were around 40 $Wm^{-2}$ in the rest of the analysed LSMs, where normally the grass surfaces were associated with improvements. Although it is not shown in this work, we detected a higher sensitivity

of this LSM to the soil type in comparison with the other LSMs. Thus, the parameters associated with clay loam could be inappropriate for this LSM, leading to a wrong partitioning of the net radiation in the model (too low Le and too high SH). In contrast, the sensitivity of RUC to the LC categories was the lowest one among the compared LSMs. In any case, the analysis of the relative contribution of the soil type on the surface fluxes is out of the scope of this study.





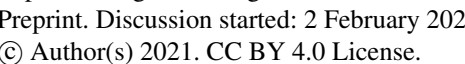

**Figure 6.** Same as in Fig. 3 but for RUC





### 3.1.5 NEW-LC experiment overview

In general, the simulation of the fluxes in the NEW-LC experiment improved over those pixels mostly covered by grass and crop. However, the specific analysis from previous subsections revealed how changing the LC representation in the model towards a more realistic one (NEW-LC) did not necessarily lead to an improvement of the fluxes in the whole analysed area for all the LSMs. This was mainly caused by the errors found in those pixels mostly covered by coniferous forest (ENF) in some of the LSMs, where the simulated values of the fluxes were extreme. This impacted notably the RMSE values over the

analysed area due to the high percentage of pixels where the dominant LC was forest (46%). This issue was more important in Noah-MP and CLM4, with overly positive SH bias and negative Le bias in conifer pixels. It should be noted that these LSMs include more and different parameters associated with this vegetation type, an issue that will be investigated later in the FOREST experiment. On the contrary, Noah is the LSM least affected by errors in the conifer forest, and, indeed, it is the LSM that most improves the simulation of the surface fluxes in the NEW-LC experiment.

This is well observed in Fig. 7, where the SH and Le at 12:00 UTC for the different dominant LC categories (colours) are plotted against the fraction covered by the dominant LC category within each pixel. Panels a to d (SH) and f to i (Le) show the simulated values for the different LSMs while panels e and j show the comparison with the observed values from the AAF. The dependency on the fraction of the dominant-category is well observed for the AAF in Fig. 7e (SH) for forest and grass pixels: the more the percentage of forest, the higher the SH, since forests were associated with the highest values of observed SH.

The opposite is observed for the grass pixels, with SH diminishing for increasing percentages since grass EC measurements showed the lowest SH values. These dependencies are slightly observed for the Le due to the similar Le values measured over all the surface types with the EC towers (see Fig. 1b and d). Coming back to Fig. 7, one can deduce that Noah is the LSM that provides the most realistic values in comparison with the AAF from a visual comparison, especially for Le.

The other LC category associated with large biases was the urban one. In this case, the effect on the RMSE of the total area

was minimum, since only the 2.4% of the pixels have urban as dominant LC category. It should be also noted that the fluxes over the urban surfaces for the AAF computation (used as benchmark) were approximated with a simple model (see Section 2.1.2), due to a lack of measurements. Hence, a fair discussion about the absolute values of the biases found over this type of LC was not possible since relatively high uncertainty also exists in the values used to evaluate it. However, the fluxes simulated by the LSMs were also too extreme, with values close to 0 $\mathrm{Wm^{-2}}$ for Le in the case of Noah and Noah-MP (see grey circles in

Fig. 7f and g), and with surprisingly low SH and high Le for the case of CLM4 (Fig. 7c and h).

Furthermore, the dependence of the AAF with the percentage of coverage of the dominant LC in each pixel is well observed in Fig. 7e and j, which is inherent to the methodology used in the AAF calculation. However, this dependence is hardly seen from the model outputs (as expected using the dominant approach, note the small slope of the scatter plots for each coloured category). As commented before, it should be noted that Noah, Noah-MP and RUC use a *dominant* approach for each grid

cell, meaning that the LSM uses some surface information (LAI, roughness length, or albedo) only from the dominant LC (Li et al., 2013). This led to well-differentiated fluxes values for each category (Fig. 7). The case of CLM4 is slightly different since it uses a 5-class sub-grid approach that separates the model grid surface into glaciers, lakes, wetland, urban and vegetated





surfaces, with up to four plant functional types for vegetated areas within the cell (Oleson et al., 2010). However, despite this tile-method that resembles a mosaic approach, the differences due to the LC categories are also well observed in the CLM4

fluxes (see Fig. 7c and h), and a clear slope in the scatter plots is not observed either for the different LC categories.

Hence, from the NEW-LC experiment, we can conclude that there is a high dependence of the fluxes on the LC type in WRF, which can lead to important biases if the parameters and processes associated with specific categories (conifer forest in this case) are not appropriate. These results agree with those found by Couvreux et al. (2016), where twelve IOPs of the same field campaign were simulated with ARPEGE (Courtier and Geleyn, 1988), AROME (Seity et al., 2011) and ECMWF models

(Simmons et al., 1989), with grid sizes of 10 km, 2.5 km and 16 km, respectively. In this work, the SH was systematically overestimated over the area analysed by the three models. Indeed, a larger SH overestimation was found for the ARPEGE model in two of the three evaluated pixels close to the area, which surface was represented as forest. This also highlights the issues found over this type of vegetation cover, which was enhanced with a *dominant* approach (as in WRF in this work and in AROME in Couvreux et al. (2016)). This simplified approach does not take advantage of the available sub-grid surface

information and motivated the MOSAIC experiment, where the sub-grid heterogeneity was taken into account in the Noah and RUC models.





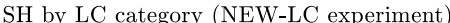

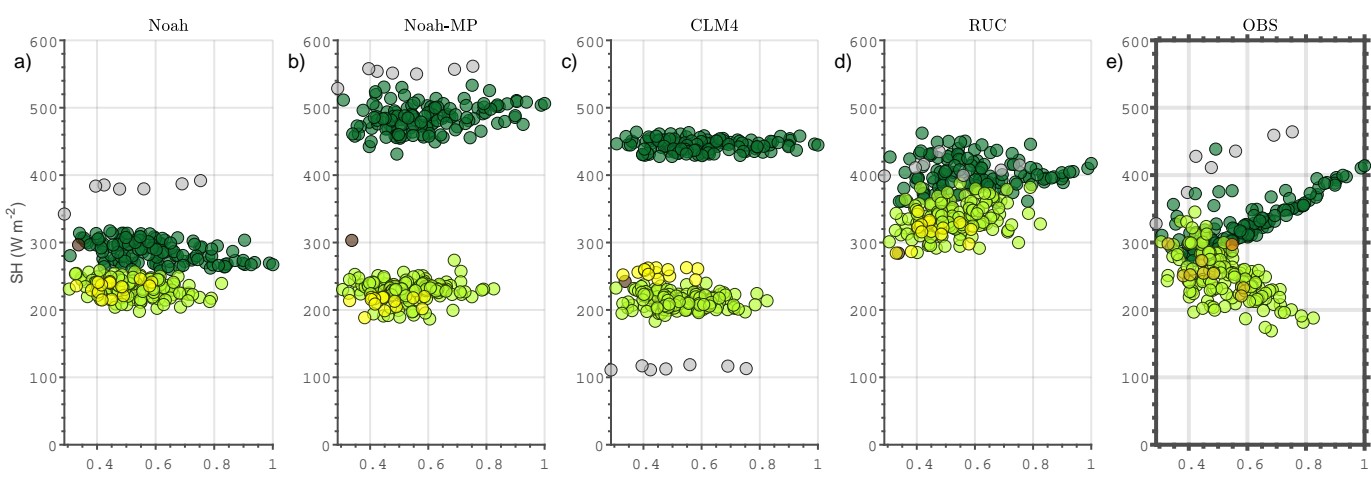

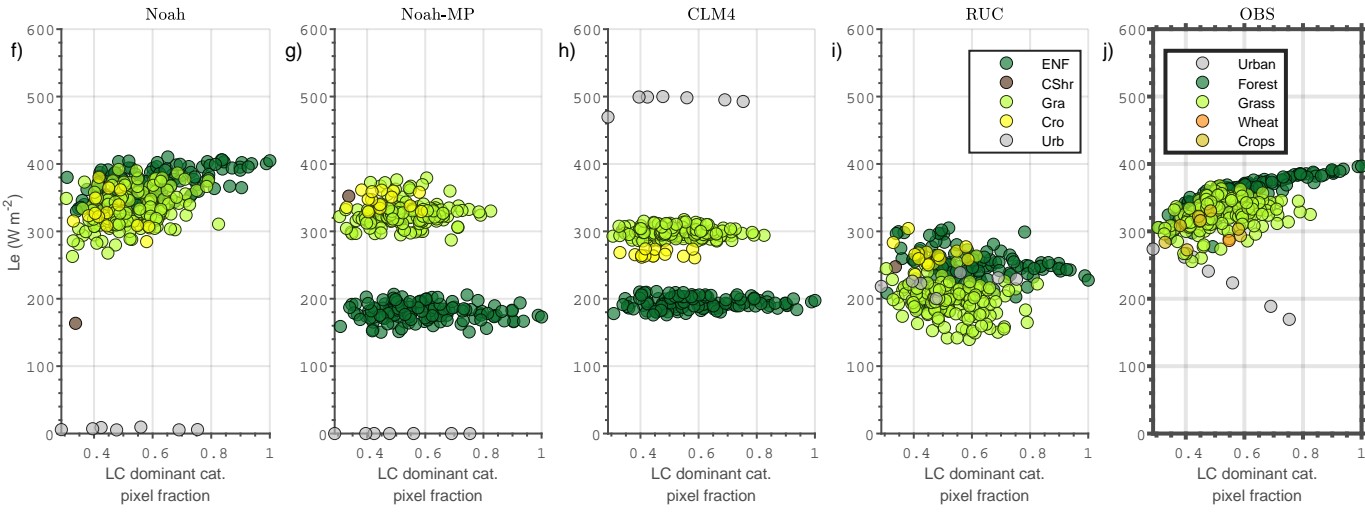

**Figure 7.** SH (upper figures) and Le (bottom figures) simulated at 12:00 UTC by each LSM: Noah (a and f), Noah-MP (b and g), CLM4 (c and h) and RUC (d and i). The x-axis shows the total fraction covered by the dominant LC category in each respective pixel. The results are provided for the different LC categories with colours. Panels e and j represent the same but for the gridded area-averaged fluxes (AAF) calculated from the EC measurements, which are used for the model evaluation.



## 3.2 MOSAIC experiment

The changes in RMSE obtained after using the mosaic approach in Noah and RUC LSMs are shown in Table 4, with significant improvements in Le and especially for the case of Noah, with a reduction of almost 20 $\mathrm{Wm^{-2}}$ in the Le-RMSE (from 63 $\mathrm{Wm^{-2}}$

to 45 $\mathrm{Wm^{-2}}$). For the case of Noah, Fig. 8 shows the simulated SH and Le obtained from the NEW-LC and the MOSAIC experiments.

The use of the *mosaic* approach (panels b and e) caused a *merging* effect among values from different categories on the simulated fluxes in comparison with the *dominant* one (panels a and d), with results that better resemble to those figures obtained from the AAF (panels c and f). This is particularly clear for SH, where the stronger dependence of the fluxes with

the type of dominant LC (Fig. 8a) is removed in the mosaic approach (Fig. 8b). The impact was larger on those pixels where the dominant LC was forest or urban, which were associated with more extreme fluxes values. Note the near zero values of Le in the dominant approach in urban pixels (panel d) and the Le values that depend on the fraction of urban within each pixel (panel e). The correlations of the fraction of coverage of the dominant LC category and the surface fluxes are included in the legend for each LC category in Fig. 8. This correlation depends on the strength of the fluxes values associated with the

specific categories. For example, the anti-correlation observed in Fig. 8e for the urban pixels is due to the strong impact when increasing the percentage of cover of this LC category. Thus, if a pixel with 100% urban were present, its Le value would tend to 0 $\mathrm{Wm^{-2}}$ using a mosaic approach, as observed when using the dominant method in Fig. 8d.

However, this *merging* effect was only slightly observed for the case of the MOSAIC experiment in RUC (Fig. 9). This was probably caused by the fact that the mosaic approach used in RUC is not applied for the albedo values, contrary to the

MOSAIC-Noah simulation. In the case of RUC, only average emissivity, LAI and roughness length are used based on the percentage of each surface. Average albedo, which is the parameter that has the highest impact on the net radiation of each grid cell, subsequently affecting SH and Le, is not used

Figure 10 shows the albedo differences between NEW-LC and MOSAIC used by the Noah (upper figures) and RUC models (bottom figures). While the values for MOSAIC-Noah (Fig. 10b) consisted of a weighted average from the different LC of each

grid cell, it was not the case for MOSAIC-RUC (Fig. 10d), which diminished the impact of the mosaic approach application in the RUC model.

It should be noted that adding a mosaic approach caused a change in the percentage technically used for some LC categories. This is well observed in the comparison of the percentages shown in Fig. 1b and d. For example, 2.2% of the pixels were characterised as urban with the dominant approach (panel d) while the coverage using a mosaic approach increased up to

9% (panel b). In this case, the mosaic approach increases the percentage of urban fluxes contributing to the averaged values, although in a much more diffuse way (concentrated in more pixels). However, the extreme effect of those pixels fully considered as by urban are removed. This is also the case for the crop pixels, that cover 9.9% of the area (wheat and corn) shown in Fig. 1b (taking into account the subgrid variability), but only 3.9% as dominant category in 1x1 km pixels (Fig. 1d).

In any case, the mosaic approach might be always more realistic than the dominant one; in the latter, the sub-grid information

is not used and a unique LC type is defined, even when the combination of secondary and similar LC categories have a higher

percentage than the dominant one (e.g., 40% of conifer, 30% of dense shrub and 30% of open shrub). If the resulting dominant category is associated with inappropriate parameters (e.g. conifer in the example and in our case), the error will be greater in the dominant approach. For this reason, it is also crucial to be cautious with some LC definitions that could lead to too extreme fluxes. This is also important for the urban pixels, where the fluxes were quite different and sometimes too extreme

depending on the LSM used. This can have an important impact on the results of the simulations, either with a dominant or a mosaic approach, as stated in Mallard and Spero (2019). Furthermore, appropriate fluxes measurements for model evaluation over urban surfaces still remain a challenge.

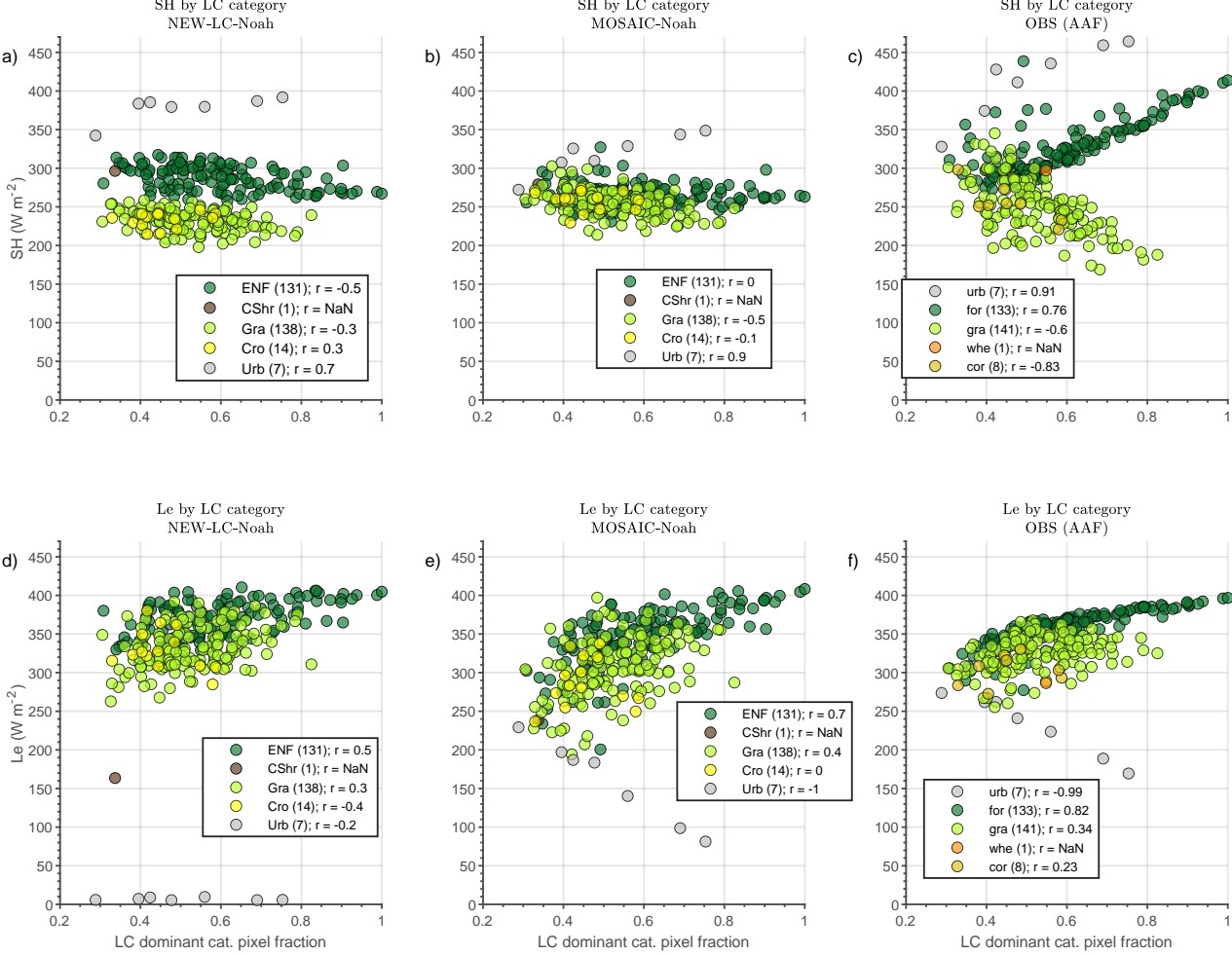

**Figure 8.** Same as in Fig. 7 but comparing the results from Noah-NEW-LC (using a dominant approach) and from Noah-MOSAIC (using a mosaic approach). Panels c and f corresponds to the area-averaged fluxes (AAF) from observations, considered as the reference. The correlation between the fluxes and the fraction of coverage of the dominant LC category is included in the legend for each LC category within each pixel, with the number of pixels used in brackets.



**Figure 9.** Same as in Fig. 8 but for NEW-LC-RUC (dominant) and MOSAIC-RUC (mosaic).





**Figure 10.** Model albedo over the analysed area. Comparison of the grid values between the dominant (left) and the mosaic (right) approaches for Noah (a, b) and for RUC (c, d). Note the absence of change in the mosaic approach using RUC.



## 3.3 FOREST experiment

The previous experiments revealed the large biases in simulated surface fluxes associated with conifer forests (ENF) for some
LSMs (Noah-MP and CLM4). These models simulated too low values of Le and too high SH in comparison with the observed
values (Fig. 7). This was also the case for RUC (Fig. 7d and i), but in this case the biases were not only limited to the ENF
pixels but to all the categories.

   Since the biases were only observed in ENF for Noah-MP and CLM4, we hypothesised that this was caused by a too
high resistance to the transpiration parameterised for these type of trees. We explained in Section 2.3.4 the motivation for
this experiment and we justified the changes applied, based on the parameter values used in Bonan et al. (2014) for ENF
forest. Specifically, we changed the values of the $g_1$, $g_0$ and $V_{cmax25}$ parameters. These modifications should lead to increased
transpiration, and therefore, also to decreased SH. Indeed, this is observed in the results, the RMSE decreases to 77 $\mathrm{Wm^{-2}}$ for
SH and to 56 $\mathrm{Wm^{-2}}$ for Le, which are lower errors compared to the NEW-LC-Noah-MP experiment (see Table 4). The results
are consistent with the fluxes shown in Fig. 11, where the SH and the Le observed at the forest site (black line) were compared
to the simulated values obtained with the central pixel of the domain completely (100%) covered by ENF (conifer, blue), or
by ENF with these parameters changed (red line). These two additional simulations shown in Fig. 11 would correspond to
the NEW-LC Noah-MP and the FOREST Noah-MP experiments, respectively. However, in these graphics the effect of the
change is analysed in a clearer way for the whole diurnal cycle. Again, it is demonstrated how the tuning of these parameters
towards less resistance to transpiration provided better results than the original parameters tabulated for ENF, both for SH
and Le. Specially, the slope of the Ball-Berry equation ($g_1$) is the parameter that most influenced the results, which does
not seem appropriate in this area for this type of trees, since it limits transpiration too much. As asserted in Medlyn et al.
(2011), $g_1$ is the key parameter for plant stomatal conductance, being quite variable among species in areas with different
environmental conditions. The authors suggested that $g_1$ should increase with increasing temperatures, as might be the case
in our study, compared to the values tabulated for this LC category in the model. The results of the present work indicate
that the tabulated values, which were developed for other region/dates or even with a different type of conifer or density of
trees, should be revised. This is consistent with the scientific demand for systematic experimental observations representative
of different scales (Vilà-Guerau de Arellano et al., 2020), from the leaf-level (as stomatal conductance) to the landscape and
model grid-scales (for example reliable area-averaged fluxes).



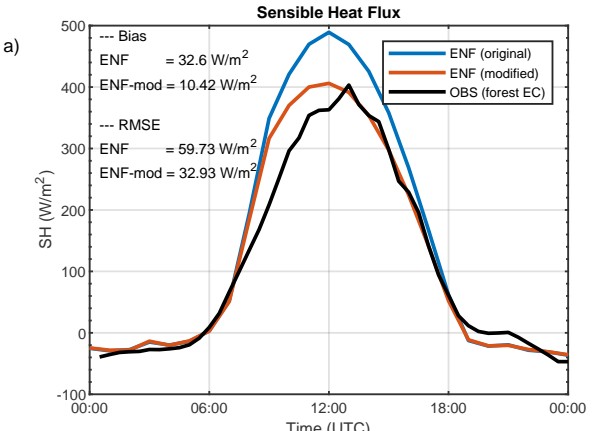
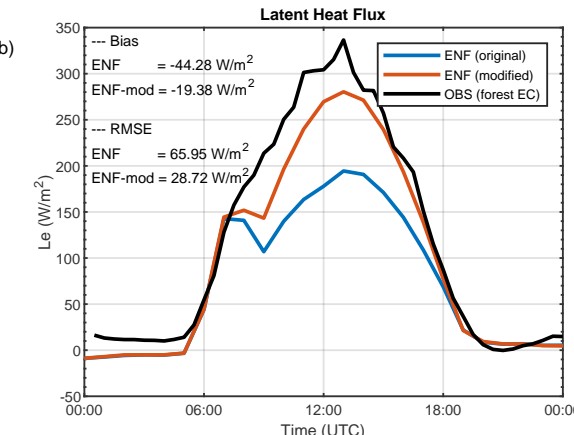

**Figure 11.** a) Sensible heat flux (SH) from two simulations with the evaluated pixel completely covered by the original ENF (conifer, blue) and ENF-mod (conifer with parameters modified, red). The observations taken over the forest site are indicated with black line. b) Idem for the latent heat flux (Le). The model results were obtained using the Noah-MP land-surface model for day 19 June 2011. The biases and root-mean-square error (RMSE) are included.





## 3.4 Analysis of the simulated evaporative fraction (EF)

The previous analyses provided information about the biases and the RMSE of SH and Le independently for the different experiments and LSMs. However, these analyses did not take into account the total energy available to be partitioned into the atmospheric fluxes, which may be different depending on the surface type. Hence, in Fig. 12 we represented the evaporative fraction (EF = Le / (Le+SH)) for each LSM (panels), sorted by real LC categories of each grid cell for the different experiments performed (DEFAULT in blue, NEW-LC in red, MOSAIC in orange and FOREST in green). The values obtained from the AAF

are also included as reference with dark grey colours in the four different panels. The highest observed values were observed in pixels with grass as dominant LC, while the lowest ones were observed in the urban pixels, followed by the forest ones, where the EF was around 0.5. The values observed from the EC towers (representing an homogeneous surface) are indicated with black circles and would represent what it would be expected in a pixel completely covered by each LC category. For example, EF is 0.73 for grass surfaces and 0.45 for forest ones.

In Noah (Fig. 12a), the DEFAULT experiment (blue) provided a large variation in the EF of each real LC category. This resulted from the unrealistic and varied LC representation in this experiment. The NEW-LC experiment (red) partially corrected the large EF variability, especially for the grass and forest pixels. Therefore, in general, not only were the SH and Le biases corrected, but also the fluxes partitioning, except for the moor and urban pixels, with scarce presence. Besides, the MOSAIC experiment (in orange) served to improve the simulated values of EF, which were very close to those obtained from the AAF.

The DEFAULT experiment in Noah-MP (Fig. 12b) systematically underestimated the EF (blue) in comparison to the observations (dark grey). The NEW-LC experiment (red) served to correct this underestimation in all LC categories except in the forest and urban grid cells. In the urban pixels, the Le was very close to 0 $\mathrm{Wm^{-2}}$, leading to very low EF. As discussed before, the Le was significantly underestimated in the forest (ENF) pixels, while the SH was overestimated, which led to underestimation of EF. This was partially corrected in the FOREST experiment (represented with green in Fig. 12b) by applying modified

parameters that made easier the transpiration from these plants. However, even with these changes, the EF values obtained from the model were far from those from the observations.

The DEFAULT experiment in CLM4 (Fig. 12c in blue) showed EF values more similar to those observed. The NEW-LC experiment produced EF values even more similar to the observations (Fig. 12c in red), except for those pixels covered by forest or urban. For the forest, the NEW-LC had the same issue as Noah-MP (i.e., too low EF associated with too low Le and

too high SH), with an important influence on the total RMSE, worsening the values (Table 4). For the urban grid cells, EF values of 0.8 were observed, meaning that an important amount of the available energy was used for Le, which was unexpected and far from the AAF calculations.

RUC (Fig. 12d) exhibited the smallest sensitivity to the modifications performed in each experiment in comparison to the other LSMs. As commented before, the LC changes in RUC had a lower impact on the results, which where more impacted

by the soil type (not shown) than by LC. Hence, RUC showed a general tendency to underestimate EF with systematic Le underestimation and SH overestimation for all the LC categories, as it was previously observed.





**Figure 12.** Evaporative fraction (EF = Le/(SH+Le)) for the different LSMs used: a) Noah; b) Noah-MP; c) CLM4; and d) RUC. The results are ordered by real dominant LC type of the pixels (vertical subdivisions) and for the different experiments performed: DEFAULT (blue), NEW-LC (red), MOSAIC (orange, only in Noah and RUC), and FOREST (green, only in Noah-MP). The values obtained from the observations (AAF) are indicated with dark-grey boxplots. Black circles indicate the EF obtained from the EC towers for each corresponding (homogeneous) surface.



# 4 Future Research: Impacts on the PBL

The results from the previous subsections revealed substantial impacts of LC on simulated fluxes and their partitioning, especially by some LSMs that simulated extreme values for specific LC categories. The impacts on the surface fluxes will also

affect the associated atmospheric variables close to the surface, with a potential influence on the full development of the PBL (Sühring et al., 2014). Hence, we think that the next step of this investigation will be to analyse the relative contribution of the changes in the surface representation of the model on the atmospheric variables within the whole PBL.

In this context, the evaluation of the simulated PBL height (zi) can be considered as a first indicator of the model ability to simulate the PBL structure. For our case study, Fig. 13 shows the simulated zi versus the simulated EF for each grid cell at

midday of day 19 of June, coloured differently depending on the LC category. The zi simulated values have been normalised by the observed zi ($zi_{obs}$, 877 m agl). This value was averaged from two radiosoundings launched at two sites of the campaign, with values of 836 m at 11:00 UTC in Site 1 (central site of the domain) and 919 m at 13:00 UTC in Site 2 (4 km to the south of Site 1). The results were provided for the 4 LSMs configured with the NEW-LC experiment, which coincided with the actual LC in the area. The squared symbols indicate the $zi/zi_{obs}$ values estimated observationally using different instruments at the

two commented sites (Table 1) and were used to evaluate the model ability simulating this variable. The EF values for these observations are those obtained from the AAF in each respective pixel (Site 1 and Site 2) at the times of the zi measurements from each instrument (slightly different, but within the time range from 11:00 UTC to 13:00 UTC, see Table 1). In general, the observations indicated a zi between 823 and 1100 m agl, depending on the instrument used to estimate this value. These values are within the range of variation of those simulated by WRF, which indicates a good performance of the model for the 4 LSMs

used, especially if the comparison is done with the modelled values in the pixels where the two sites were located (red and blue circles in Fig. 13 for Site 1 and Site 2, respectively).

For the case of Noah LSM (Fig. 13a), the simulation of zi did not show a clear dependence of zi on EF or on the LC type, fitting well with the observed values for most of the surfaces, with zi values ranging within the same range of variability observed from different instruments. This was also the case for the values simulated by Noah-MP (Fig. 13b) and CLM4 (Fig.

13c) over the grass pixels. However, these LSMs also showed a higher variability in the simulated zi depending on EF and on the type of LC. Conifer surfaces, which were associated with EF underestimation due to SH overestimation, systematically simulated higher values of zi. The same was observed for the urban pixels in Noah-MP (with extremely low EF values), while CLM4 simulated the lowest zi values over the urban surfaces, due to the very high simulated EF (low SH). For the case of RUC (Fig. 13d), the dependence of zi on EF or LC category was smaller, but this LSM tended to provide slightly higher values of

zi, also associated with lower values of EF (SH was systematically overestimated and Le underestimated).

The results shown in Fig. 13 were those obtained from the NEW-LC experiment, which are summarised in Table 5 in the form of mean $zi/zi_{obs}$ values for each LSM and for each LC category. This table also include those results obtained with the MOSAIC experiment for Noah and RUC (indicated with brackets). Indeed, the effect of using the mosaic approach on zi is not very significant, with small changes that did not changed significantly the PBL height. The MOSAIC experiment integrates

the small-scale heterogeneities of less than 1 km, which are not expected to significantly impact the zi. In any case, these small



**Figure 13.** Height of the planetary boundary layer ($z_i$) simulated by WRF normalized by the $z_i$ observed from radiosoundings* ($z_{i_{obs}}$) versus the evaporative fraction (EF) simulated at 12:00 UTC of 19 June 2011 in each pixel for the whole analysed area. The results are separated by LC categories (circles with colours). The simulated values obtained over the pixels of Site 1 and Site 2 of the BLLAST campaign are indicated with red and blue circles, respectively. Results are shown separately for the 4 LSMs analysed using the NEW-LC experiment: a) Noah; b) Noah-MP; c) CLM4; d) RUC. The $z_i/z_{i_{obs}}$ values estimated observationally are also indicated with squares for model-observation comparison; from different instruments and for Site 1 (red palette colours: radiosoundings (RS), unmaned aerial vehicles (SUMO), wind profiler (UHF) and LIDAR) and for Site 2 (blue: RS). The observational EF values are those from the AAF in the pixels of Site 1 and 2 at the time of the $z_i$ measurements by each instrument (around 12:00 UTC ±1 hour). *The $z_i$ used for the normalization (877 m) corresponds to the mean value of $z_i$ at Site 1 at 11:00 UTC (836 m) and $z_i$ at Site 2 at 13:00 UTC (919 m).





changes tended to slightly modify $z_i$, for example diminishing $z_i$ in the case of the ENF or urban pixels, due to the reduction in SH when taking into account more LC categories and not only ENF within each pixel. In any case, the MOSAIC experiment could only be tested for Noah and RUC, where the fluxes values were not too extreme, leading to very similar $z_i/z_{i_{obs}}$ values for all the LC categories (within the range from 0.99 in CShr to 1.08 in ENF). In the case of Noah-MP and CLM4, the simulated

$z_i/z_{i_{obs}}$ values differed more among the LC categories (from 1.06 to 1.22 for Noah-MP and from 0.89 to 1.15 for CLM4). Hence, the effect of applying a MOSAIC approach would be expected to have a larger impact on these LSMs with a larger range of fluxes and $z_i$ variability. From a general analysis, the MOSAIC-Noah experiment was the one with $z_i$ values closer to the observations ($z_i/z_{i_{obs}} = 1.02$), which coincides with the general evaluation done for the surface fluxes (Table 4).

**Table 5.** $z_i/z_{i_{obs}}$ mean values simulated by WRF for each LSM (raws) and for each LC category in the model (columns). The results are those obtained with the NEW-LC experiment, which coincided with those shown in Fig. 13. The values obtained for the MOSAIC experiment (for Noah and RUC) are indicated in brackets. Results averaged for the whole area are shown in the last column.

| LSM | ENF | CShr | Gra | Cro | Urb | All area |
|---------|-------------|-------------|-------------|-------------|-------------|-------------|
| Noah | 1,08 (1,06) | 0,99 (0,86) | 1,06 (1,07) | 1,07 (1,11) | 1,04 (1,02) | 1,05 (1,02) |
| Noah-MP | 1,18 | 1,06 | 1,07 | 1,06 | 1,22 | 1,12 |
| CLM4 | 1,15 | 1,05 | 1,08 | 1,09 | 0,89 | 1,05 |
| RUC | 1,17 (1,18) | 1,11 (1,19) | 1,16 (1,18) | 1,14 (1,17) | 1,19 (1,22) | 1,15 (1,19) |

These results indicate how the important effects of the LC type on the surface fluxes in the case of some LSMs (and especially

for some LC categories) are transferred to the top of the PBL, affecting $z_i$ even from an analysis of this variable at a resolution of 1x1 km. Thus, we have illustrated the impacts that the surface representation have on the development of the PBL in the model, which encourages further future scientific analysis on this topic. Specifically, it would be interesting to analyse how the different scales of the heterogeneous surface patches impact the PBL development in the model (Sühring et al., 2014). That is, future work will try to answer the questions about how the scale, the type (LC categories) and distribution of the heterogeneous

patches of the surface affect the PBL development, as well as how these changes differ depending on the land-surface and PBL schemes of the model. However, this analysis deserves a fully dedicated strategy disentangling the surface effects from those from advection, subsiding meso or synoptical motions (e.g., Pietersen et al., 2015) and entertainment processes at the top of the PBL (e.g., Garcia-Carreras et al., 2015), as well as extensive $z_i$ measurements at different points of the area analysed will be needed for the evaluation. This future research will help to understand how the surface forcing affects the PBL and to what

extent the processes reproduced in the model differ from those observed in the reality.





## 5 Summary and conclusions

The changes in the LC of the Earth's surface trigger varied and sometimes unpredictable consequences at different spatio-temporal scales, affecting biophysical processes in the soil and in the atmosphere. Hence, it is crucial to know more about the impacts of the LC changes on all these processes. In this work, we investigated the sensitivity of turbulent heat fluxes simulated
by the WRF model to the manner in which the surface is represented in it.

To this aim, different sensitivity experiments were performed for a case study over an heterogeneous area in the south of France. They were evaluated with gridded area-averaged fluxes (AAF), computed from tower measurements installed over five vegetation types (forest, corn, wheat, grass and moor) during the BLLAST field campaign. In order to add robustness to the study and to detect differences, the experiments were carried out using four LSMs available in WRF: Noah, Noah-MP, CLM4
and RUC.

First, a control experiment was performed with the default options in WRF: LC from the IGBP-MODIS database and a dominant sub-grid approach, i.e., the model used the tabulated surface parameters of the LC category with the highest percentage of coverage in each pixel. We hypothesised that these simulations were limited because of the large differences between the LC representation and the actual surface, and because of the loosing of LC information at the sub-grid scale.

Thus, a new experiment was designed (named NEW-LC), which improved the surface representation by adding the LC information from the 30-m resolution CESBIO maps, which were much more accurate and realistic in the area than the default IGBP-MODIS dataset. The observed changes in the surface fluxes were dependent on the LSM used, due to their differences in the parameters associated with each vegetation type, and also to their different representation of the surface processes. The improvement was clear for Noah for all the LC categories. RUC was the LSM that showed the weakest response of the fluxes
to the LC categories, without substantial changes in the scores. Noah-MP and CLM4 showed some improvements in those pixels covered by crops or grass, but they also exhibited an important SH overestimation and Le underestimation in the surface fluxes simulated over those pixels mainly covered by conifer forest (ENF). The ENF biases contributed significantly to the total model error due to their relatively high percentage of coverage (45%) in the analysed area. The NEW-LC experiment revealed the need for a correct representation of LC in the analysed area, in part due to the high dependency of the fluxes on
the LC categories. In addition, the appropriate characterization of surface parameters associated with some LC categories (e.g., conifer) still needs to be improved, as it was also discussed in previous works (e.g., Li et al., 2013; Cuntz et al., 2016).

In the second experiment (named MOSAIC), the sub-grid heterogeneity (below 1 km) was taken into account with a mosaic approach in Noah and RUC, meaning that the fluxes in each grid were calculated as weighted averages from individual surface fluxes obtained from each tile or LC category. The mosaic approach caused more homogeneity among the surface fluxes
simulated in the analysed pixels, which corresponded better to the AAF used as benchmark data. This improvement in the surface representation led to improved scores for Noah (especially for Le) while smaller changes were observed in RUC. This was because in RUC the mosaic approach did not include the use of pixel-averaged albedo values based on the percentages of each LC category, as done for the surface roughness, LAI and emissivity. In Noah, the albedo was also averaged, significantly





contributing to the improvements, since the albedo is the parameter that seems to have a larger impact on the net radiation
available to be partitioned into SH and Le.

Finally, a last experiment (named FOREST) was motivated by the issues found in the conifer pixels for some LSMs (especially Noah-MP and CLM4). The modifications in the FOREST experiment were conducted to reduce the resistance of conifer trees to transpiration, using updated parameters as used in Bonan et al. (2014). The effect of these changes was to facilitate the transpiration processes, coinciding better with the observations and improving the scores.

This work demonstrates again the importance of a correct representation of LC in the area which is evaluated, as it was also shown in previous works (Cheng et al., 2013; Schicker et al., 2016; Jiménez-Esteve et al., 2018). This can considerably affect the simulation of the fluxes that will drive the associated boundary-layer processes, as discussed in the last section, where the study of the impacts on the PBL is encouraged as future research. Besides, it is worth using a mosaic approach to benefit from the sub-grid surface information that is normally available.

In any case, the particular conditions of the region analysed make that some specific conclusions might not be applicable to other regions. For example, it is also possible that in our case we observed more ET over the conifer trees than in the model due to the possible particularities of the area. Hence, the parameters adjusted in the model for the conifers could be due to the differences among species belonging to the same LC category (Granier et al., 1989), tree density, tree age (Sellin, 2001), or to the specific surface conditions of this case study (for example with relatively high values of SM). All these aspects open an
interesting new line of research with the objective of improving the parameters associated with each vegetation type, which could be achieved by including leaf-level measurements of stomatal conductance in experimental campaigns, as recently stated in Vilà-Guerau de Arellano et al. (2020).

The possible uncertainties in the EC measurements used to evaluate the model should be also taken into account, especially over those vegetation types where it is, somehow, more difficult to have accurate or representative high-frequency measure-
ments, as in the case of the forest. Furthermore, the calculation of the AAF consists on important assumptions based on the spatial extrapolation of EC data that can add uncertainty to the data used for the evaluation. In our case, they were also constructed using a simple estimation of the fluxes for urban surfaces, due to the lack of measurements. This could be also associated with errors (although with a small percentage of coverage in the analysed area). All these necessary simplifications that were done highlight again the importance of having extensive measurements over a wide variety of surface types (Cuxart and Boone,
2020) and including atmospheric, soil measurements, and those related to the plant physiology and status, even with field strategies more ambitious than the BLLAST campaign, which already included a quite important deployment of instruments. This will help to improve the evaluation process of models, a needed step to continue advancing in their development.





## Appendix A

Table A1 shows the LC transformation performed in the NEW-LC experiment.

**Table A1.** LC categories of CESBIO and respective transformation to the IGBP-MODIS LC categories. The respective codes of each dataset are included in the central columns. * All crop types were transformed to the single cropland category available in IGBP-MODIS; however, most of the crop types in the area were summer crop (mainly corn). ** Deciduous forests were transformed to the LC category of evergreen needleleaf forest (conifers), even when a deciduous broadleaf forest category is available in IGBP-MODIS. This was done due to the lack of measurements over deciduous trees, which made that the area-averaged maps used to evaluate the model were constructed with data from the conifers; this strategy allowed a fairer model-observation comparison. Abbreviations used along the article in the text and figures are indicated with brackets.

| Name and code in CESBIO | | Code and name in IGBP-MODIS | |
|---|---|---|---|
| Summer crop* | 11 | 12 | Cropland (Cro) |
| Winter crop | 12 | 12 | Cropland (Cro) |
| Deciduous broadleaf forest** | 31 | 1 | Evergreen needleleaf forest (ENF) |
| Evergreen needleleaf forest | 32 | 1 | Evergreen needleleaf forest (ENF) |
| Grass | 34 | 10 | Grassland (Gra) |
| Moor | 36 | 6 | Closed shrublands (CShr) |
| Dense urban | 41 | 13 | Urban (Urb) |
| Diffuse urban | 42 | 13 | Urban (Urb) |
| Industrial and commercial areas | 43 | 13 | Urban (Urb) |
| Roads | 44 | 13 | Urban (Urb) |
| Mineral surfaces | 45 | 16 | Barren or sparsely vegetated |
| Beaches and dunes | 46 | 16 | Barren or sparsely vegetated |
| Water | 51 | 17 | Water |
| Glaciers and snow | 53 | 15 | Snow and ice |
| Prairies | 211 | 10 | Grassland (Gra) |
| Orchards | 221 | 8 | Woody savanna (WSa) |
| Vineyards | 222 | 8 | Woody savanna (WSa) |




## Appendix B: Technical details about the experiments

**NEW-LC experiment**

Two variables were modified in the *geo_em_d04.nc* file (the fourth-domain output from the *geogrid.exe* program of the WRF preprocessing system (WPS)). On the one hand, the *LANDUSEF* variable was modified, including the new percentages for each LC category in all the 1-km grid cells of the fourth domain. The same was done for the rest of the domains, but only in the area covered by the inner domain. On the other hand, the dominant LC category in each pixel was calculated based on the new information, which served to modify the *LU_INDEX* variable of the same files than before.

These files with the modified *LANDUSEF* and *LU_INDEX* variables were re-incorporated to the WPS system and the rest of the preprocessing programs were executed, i.e., *ungrib.exe* and *metgrid.exe*, obtaining the final *met_em* files used to run the model. Note how in order to use these modified files in the model simulations, the *surface_input_source* parameter in the WRF *namelist.input* file was set to 3.

**MOSAIC experiment**

To activate the mosaic approach, some options should be included in the *namelist.input* file of WRF. In the case of Noah, the *sf_surface_mosaic* option should be set to 1. Besides, the *mosaic_cat* option indicates the maximum number of tiles to be used, which was set to 19, the maximum possible in our area. In the case of RUC, the *mosaic_lu* and *mosaic_soil* should be included and set to 1.

**FOREST experiment**

The three parameters modified in the FOREST experiment were changed in the *MPTABLE.TBL* file of WRF. This file contains the vegetation parameters tabulated of the different LC categories for the two LC datasets available in WRF. Since our simulations used the IGBP-MODIS LC dataset, we changed these parameters in its corresponding section within the file. Specifically, we modified *MP* from its original value (6) to 9, *BP* from 0.002 to 0.01 $mol\,H_2O\,m^{-2}s^{-1}$, and $V_{cmax25}$ from 50 to 62.5 $\mu mol\,m^{-2}s^{-1}$ for the LC type 1 (ENF). This was done before running the WRF simulation with the Noah-MP LSM.



*Code and data availability.* The source code of the Weather Research and Forecasting model (WRF v4.1.3) is available at https://github.com/wrf-model/WRF/releases (last access: January 2021). The initial and boundary data used for the specific period analysed can be downloaded at https://rda.ucar.edu/datasets/ds083.2/ (last access: January 2021). The CESBIO land used dataset for 2011 can be downloaded at the OSO

CESBIO webpage: http://osr-cesbio.ups-tlse.fr/ oso/posts/2016-10-06-cartes-2009-2011/ (last access: January 2021). The BLLAST data are accesible at https://www7.obs-mip.fr/bllast/. All the data and scripts used in this work are available in Zenodo (Román-Cascón et al., 2021) (https://doi.org/10.5281/zenodo.4449761), including: 1) The area-averaged fluxes (AAF) data used to evaluate the model, including data and scripts; 2) The scripts and data used to prepare the WRF experiments, including the modified geo_em*.nc files used to change the land cover of the domains; 3) The scripts and data used to process and analyse the WRF output; 4) The scripts used to prepare the figures, and; 5) The

WRF output for the used domain for each simulation used in this work.

*Author contributions.* C. Román-Cascón carried out the main analyses and wrote the article. Marie Lothon and Fabienne Lohou supervised the main scientific strategy of the work, as well as they organised the BLLAST field campaign. Oscar Hartogensis, Jordi Vila-Guerau de Arellano and David Pino originally developed the area-averaged fluxes used for fluxes computation and contributed to the scientific discussion of the work. Carlos Yagüe and Eric Pardyjak contributed to the field campaign and to the scientific discussion of the work. All

the authors contributed to the improvement of the manuscript text and participated in the BLLAST field campaign. All authors have read and agreed to the published version of the manuscript.

*Competing interests.* The authors declare that they have no conflicts of interest.

*Acknowledgements.* Carlos Román-Cascón work was funded through a Postdoctoral Grant from the Centre National d'Études Spatiales (CNES). The BLLAST field experiment was made possible thanks to the contribution of several institutions and supports: INSU-CNRS (In-

stitut National des Sciences de l'Univers, Centre national de la Recherche Scientifique, LEFE-IDAO program), Météo-France, Observatoire Midi-Pyrénées (University of Toulouse), EUFAR (EUropean Facility for Airborne Research) and COST ES0802 (European Cooperation in the field of Scientific and Technical). The field experiment would not have occurred without the contribution of all participating European and American research groups, which all have contributed in a significant amount. BLLAST field experiment was hosted by the instrumented site of Centre de Recherches Atmosphériques, Lannemezan, France (Observatoire Midi-Pyrénées, Laboratoire d'Aérologie). The

60m tower is partly supported by the POCTEFA/FLUXPYR European program. BLLAST data are managed by SEDOO, from Observatoire Midi-Pyrénées and maintained by the French national data center Data Terra/AERIS. The French ANR (Agence Nationale de la Recherche) supported BLLAST analysis in the BLLAST_A project. The authors acknowledge the support of the U.S. National Science Foundation grant number PDM-1649067 and the Spanish governments project CGL2015-65627-C3-3-R and CGL2016-75996-R (MINECO/FEDER). This work takes also part of the Modelisation and Observation of Surface/atmosphere Interaction (MOSAI) project supported by ANR and which

leans on ACTRIS-FR infrastucture Reasearch. Thanks also to the OCASO project (ref. 0223_OCASO_5_E) for the access to the server used to run the WRF model. Thanks to the National Centers for Environmental Protection (NCEP) for the NCEP FNL (final) Operational Model Global Tropospheric Analyses data, continuing from July 1999 (NCEP, 2000), which were used to initialise the WRF model. Finally,



thanks to CESBIO for the land cover data and for the soil moisture data from the DISPATCH soil-moisture product used to test some initial conditions, especially Olivier Merlin and Nitu Ojha.



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
