# Peer review of "Surface representation impacts on turbulent heat fluxes in WRF (v.4.1.3)"

_Geoscientific Model Development, 2020_

## Author Comment (AC1)

*10 March 2021*

**Response to reviewer comment on gmd-2020-371 (*Wayne Angevine, 12 Feb 2021*)**

**by Román Cascón and co-authors**

**Original comment** (Wayne Angevine, reviewer):

1. *I have concerns about the use of different land surface models with the same input data. Each LSM has its own "climate", and the input data (assuming it all comes from FNL, as implied by table 2) comes from a different model with a different climate. This is strongly demonstrated by Angevine et al. (2014) in ACP for the same project. Have the model outputs been checked to be sure that there are not strong spinup effects in the soil? I particularly suspect that this is responsible from some of the behavior of the RUC LSM, which is known to have a different soil moisture baseline from Noah.*

**Response from authors:**

Dear reviewer,

Thank you very much for all your comments. We will take all of them into account in the next version of the manuscript, which will be provided at the end of the discussion stage. However, we would like to include here a more detailed response to your comment 1, related to the soil moisture (SM) spin up for the different land surface models (LSMs).

Indeed, we do agree with your comment and we have performed different simulations including longer spin-up times to check the differences obtained (shown later).

However, we have decided to avoid performing individual spin-up simulations for all the experiments of the paper using the different LSMs due to two main reasons:

1. The first one is related to the SM spatial heterogeneity that is obtained after spinning up the model, as commented in lines 165-169 of the current manuscript:

"*Regarding the effect of possible SM differences within the area, two aspects should be noted. On the one hand, the potential SM horizontal variability due to possible inhomogeneous precedent rainfall over the area was not taken into account. However, the SM input in the model is provided with a coarse resolution of 1º and does not show any small-scale details. This is a well-known limitation of mesoscale models which is sometimes addressed through the assimilation of SM data from satellites or with previous long simulations that serve to spin-up the surface in order to obtain the appropriate SM initial values (De Rosnay et al., 2013; Angevine et al., 2014; Santanello et al., 2016). In our case, this limitation of the mesoscale modelling is an advantage because it allows to perform a fairer model-observation comparison since this limitation also exist in the AAF.*"

That is, if we decide to spin-up the model (which is more realistic), we will lose the SM spatial homogeneity that is in the SM used to initialise the model from the NCEP-FNL data (1º). As mentioned, this homogeneity is an advantage since we evaluated the model with area-averaged fluxes (AAFs) that omit the SM spatial variability, which allowed us to perform a fairer model-observations comparison. This is well observed in Fig. 1, where we compare the SM from NCEP-FNL at 5 cm used in the simulations included in the original manuscript (note how the SM is the same for Noah (a) and RUC (b)). As shown in the figure, we have

obtained a more probabilistic soil moisture field with a spin-up of 1 month (panels c and d). However, we have introduced an undesirable effect in our numerical design: soil moisture fields are now heterogeneous and therefore more difficult to be compared with AAF.

[Figure]

**Figure 1.** Soil moisture at 5-cm depth (first soil layer for Noah and the second one for RUC) valid at 12:00 UTC of 18 June 2011 in the area of interest (19x19 pixels of 1-km resolution). The central site of the BLLAST campaign is indicated with a black x. The SM is the same for all the simulations included in the original manuscript (panels a and b). Panels c and d show the SM obtained after a spin-up time of 1 month using Noah (c) and RUC (d). Note the spatial heterogeneity obtained, related to the heterogeneous previous rainfall, to the land use (see for example the urban pixels in panel c comparing to Figure 2d of the original manuscript) and, especially, to the soil type (blue box with less SM in the SW of the region (loam), in comparison with the rest of the area (clay loam, which retains more SM)). We would like to remember that, in our study, we removed this area of loam in the SW to avoid this strong effect of the soil type.

2. The second reason to avoid spinning up the model is an indirect effect. Performing long spin up simulations with different LSMs leads to SM differences caused by the differences in the WRF rainfall that are simply due to the use of different LSMs. This is well observed in Fig. 2, where we show the cumulative rainfall in each pixel after 1 month of spin-up for Noah and RUC.

[Figure]

**Figure 2.** Total cumulative rainfall (in mm) produced by Noah (a) and RUC (b) after one month of spin-up time (from 17 May to 18 June 2011). Note the spatial differences in the rainfall patches, probably caused by differences in the position and strength of simulated convective systems (typical in the region on these dates).

Therefore, we think that for the objectives of our study it is better to avoid a period with spin up since the data used for the comparison (AAF) assumes SM spatial homogeneity (point 1),

as we have in the original NCEP-FNL data. Besides, the work compares different LSMs and the application of individual spin-up periods can cause differences in rainfall among the LSMs that will affect the SM and, therefore, the fluxes (point 2).

However, we do agree that performing long spin up simulations is probably the best practice to include appropriate initial SM values in the model, which allows including a more realistic SM heterogeneity and values that agree with the dynamics (climatology) of each LSM.
* * *
Concerning the reviewer suspects about the different dynamics of RUC, we checked the differences in SM dynamics between the 4 LSMs used. Indeed, as commented by the reviewer, it seems that RUC has a remarkable different SM dynamic, which is shown in Fig. 3 for the period of the simulations analysed in the paper. For this reason, we decided to investigate these differences by performing two simulations with a spin-up time of approximately 1 month using Noah and RUC (Fig. 4).

[Figure]

**Figure 3.** SM at approximately 5 cm depth simulated by the different LSMs in the original simulations of the manuscript (without spin-up). Note the different dynamics of RUC.
.

[Figure]

**Figure 4.** SM at 5 cm (solid lines) simulated by Noah (blue) and RUC (red) in the central pixel. The SM included in the original simulations (without spin up), is indicated with a black point (~0.34 $m^3/m^3$), to be compared to the value obtained after the spin-up with Noah (~0.35 $m^3/m^3$) and with RUC (~0.38 $m^3/m^3$). Rainfall (mm/h) is included with symbols (total of 242.5 mm with Noah and 252.2 mm with RUC).

Fig. 4 demonstrates the suspects of the reviewer for the different behaviour of RUC in comparison with the other LSMs. Indeed, the SM dynamics in this LSM is different in comparison with the others and, therefore, the initialisation with the default SM initial data from NCEP-FNL is not appropriate. For the case of Noah, the value originally used, obtained from the NCEP-FNL data (~0.34 m³/m³) is quite similar to the value obtained after the spin-up period (~0.35 m³/m³). However, the differences are larger for RUC (~0.38 m³/m³).

We then consider that the initial SM at 5 cm used in RUC should be higher than the value used in Noah (according to Figures 3 and 4), being a value that must be approximately 0.03-0.04 m³/m³ higher. However, this is not the case at deeper levels, since each LSM has a different physics within the soil. Indeed, Fig. 5 shows the SM at 150 cm (for Noah, 4th level) and 160 cm (for RUC, 5th level). In this case, the SM is higher in Noah than in RUC.

[Figure]

**Figure 5.** As in Fig. 4 but for a deeper soil level: 150 cm for Noah (4th level), 160 cm for RUC (5th level).

Therefore, the SM differences between LSMs also depends on the soil levels, which makes it more difficult to apply a SM correction for RUC to initialise the model (e.g., it is not correct to apply +0.3 or +0.4 m³/m³ in all the levels to the NCEP-FNL initial SM values used in our case). Hence, we cannot easily correct the SM initial value for RUC if we want to maintain the SM horizontal homogeneity we wanted to compare with the AAF.

In any case, it is demonstrated that the superficial SM should be higher for RUC, which will act correcting the systematic underestimation of latent heat flux (Le) and overestimation of sensible heat flux (SH) observed in our study for RUC in all the land cover categories (see panels b and e of Figure 6 of the original manuscript).

We have computed the scores that would be obtained using the 1-month spin up with RUC, obtaining improvements in both surface fluxes (Table I).

**Table I.** Comparison between scores for NEW-LC-RUC with and without spin up.

|  | NEW-LC-RUC (NO spin-up) | NEW-LC-RUC (spin-up) |
|---|---|---|
| Total bias SH | 77 W/m² | 47 W/m² |
| Total bias Le | -70 W/m² | -46 W/m² |
| Total RMSE SH | 90 W/m² | 68 W/m² |
| Total RMSE Le | 85 W/m² | 68 W/m² |

Nonetheless, we do agree that the impact of the SM initialization should be, somehow, better addressed or commented in the paper. We then suggest to include in the paper an additional experiment named "SPIN-UP", commenting on this point, adding a link to this online discussion and including the scores obtained after spinning up the LSMs: Noah and RUC results have been shown here; the results for Noah-MP does not show significant differences with or without spin up; for CLM4, we have found some technical issues with the spin-up simulation, leading to unrealistic radiation values in the morning transition that affect the fluxes*.

We think that adding the information obtained from these experiments completely fits with the main objective of the paper, which is to investigate the surface representation impacts on the fluxes (including now the soil moisture initialisation).

* We continue investigating this issue with CLM4.

---

## Author Comment (AC2)

**RESPONSES TO REVIEWER #1**

**RC1**: 'Comment on gmd-2020-371', Wayne Angevine, 12 Feb 2021

The manuscript describes tests of several aspects of land surface representation in WRF with respect to heat flux from the surface. It is generally well-written. The problem is very important. The results should be useful to the many readers applying WRF to their problems. The presentation could be improved by removing some sections and clarifying others. I offer a number of general comments, but overall it is a good paper, and addressing the comments should not require doing more simulations.

The authors would like to thank Reviewer #1 for his valuable comments that have served to improve the manuscript. Please, find below (in blue) detailed answers to your comments.

General comments:

I have concerns about the use of different land surface models with the same input data. Each LSM has its own "climate", and the input data (assuming it all comes from FNL, as implied by table 2) comes from a different model with a different climate. This is strongly demonstrated by Angevine et al. (2014) in ACP for the same project. Have the model outputs been checked to be sure that there are not strong spinup effects in the soil? I particularly suspect that this is responsible from some of the behavior of the RUC LSM, which is known to have a different soil moisture baseline from Noah.

A detailed answer of this comment has been previously included in the interactive discussion (https://doi.org/10.5194/gmd-2020-371-AC1 and associated attachment). Finally, we have included these spin-up pre-experiments in the paper, which are very appropriate for the topic investigated in our study. The changes in the new version of the manuscript are included in Section 2.3.5 (line 337) and Section 3.5 (line 585).

2. The evaluation of PBL height (sections 2.1.3 and 4) is too incomplete to be useful. In such a small area, it is not possible to learn anything important by looking at individual columns, there is too much interaction between columns at any reasonable wind speed. The authors say as much at the end of section 4. The paper would be strengthened by removing these sections.

Based on the reviewer suggestion, we have removed this section in the new version of the manuscript.

3. Throughout the paper, RMSE is used as the major metric. There are two problems with this. First, RMSE includes both bias and random error. It is much more useful to treat the two separately (bias and standard deviation). Second, it is not clear how the RMSE is calculated. Please state clearly exactly what time series are being compared, including time and locations (pixels or categories).

We do agree. Now we have included a better explanation about how the different scores are computed in *Section 2.1.3 - Model Performance scores* (line 206). More information is also included in the figure caption of Figure 3.

Besides, we have included the standard deviation of the differences between the model and the observations as a measure of the random error. This is a quite interesting suggestion to complement the bias (systematic error). Now we provide the standard deviation in figures 3, 4 and 5: total std in the legend and with horizontal pointed lines and std calculated for each LC category with the symbol *. Area-averaged values are also included in the new Table 3, which is now more complete with the information of bias, std and RMSE. Note how we conserve the RMSE since we think that in some cases it is a quick and good indicator of the model performance, especially when we merge LC categories with different sign in their bias.

The text has been updated appropriately along the whole article, with an improved analysis of the different metrics (please, see document with tracked changes to see it).

4. Noah-MP is an extremely complex LSM with many configuration options, intended for use in ensemble systems. It is not clear that using the default options is correct. It might be necessary to consult the scheme developers, or to drop this option from the comparison.

We have contacted the model developers and the default options are appropriate to be used. Therefore, we have maintained this LSM in our study. Nonetheless, the model developers have also encouraged us to try the numerous options possible in Noah-MP to determine the best configuration specific to our zone. While this investigation is quite interesting, this requires a new effort with numerous simulations (probably a different study).

On the other hand, please note how we have decided to remove those simulations carried out with CLM4. This has been done due to a strange and unrealistic radiative effect found in the inner domain of WRF: after the second analysed day, there is a delay in the morning increase of SWDOWN, showing values equal to 0 $W/m^2$ until some hours after sunrise. This error also appears in subsequent analysed days, but with a delay that lasts 3 hours more every day. This possible bug leads to an unrealistic surface energy budget, affecting the fluxes.

In our case the issue only affected some hours in the morning of day 19[th] since we only analysed the second simulated day (19[th] June) from 09 to 15 UTC, but the simulation is not correct. We have tried to solve it and ask the WRF scientific community ( https://wrfforum.com/viewtopic.php?f=43&t=11824 ), but we do not have a solution at the current date. Therefore, we have decided to remove CLM4 from the paper (all the manuscript lines discussing CLM4 and its figures have been removed). Figure R1 shows this issue for a period used in the SPIN-UP simulations:

[Figure]

**Figure R1.** Short wave radiation simulated by Noah and CLM from 17 April 2011 to 22 April 2011 in the central pixel of the simulations for Noah and CLM4, illustrating the issue found in CLM4 (not solved at the present day) and affecting the total radiation available at the surface when several days are simulated. Note how this is observed only in the inner domain (4th domain of 1 km of resolution) and in land pixels (in lake pixels the radiation seems normal). We have tried to change different options in the WRF namelist.input file without success.

5. The urban land class has large biases. This is a known problem with the LSMs in WRF when run without an urban parameterization. Basically the LSM treats the urban class as a slab of concrete with no moisture availability. Since urban parameterization is out of the scope of the paper, and the AAF comparison is problematic for this category, I recommend ignoring the urban class except for a brief mention.

We do agree, indeed, we tried to avoid a full discussion about urban in the work, but following your advice, we have removed more text about the discussion of the urban class in the new version of the manuscript. We have also included at the beginning of the paper (lines 251 to 257) that the urban analysis is out of the scope of this study for the commented reasons.

However, it is difficult to effectively remove the effect of the urban class from our analysis. We could remove the analysis in those pixels where the urban class is dominant (8 pixels, see Fig. 2d). However, the urban class is also present in other pixels as a sub-category (not dominant), as seen in Fig. 2b, and this is considered in the AAFs calculation (and in the model mosaic approach).

Therefore, we maintain the results obtained for this class in the figures to illustrate the associated issues (which can impact the scores), but we have limited the discussion. Besides, the urban class has interesting effects in the MOSAIC experiment, where it is shown how important is the treatment of the sub-grid heterogeneity (including urban).

6. In the mosaic approach, I would have expected that soil properties depend on the tile class, not just radiative properties. Is this not the case? This arises in line 510 discussing RUC, but we need to know what is varied in Noah also.

The mosaic approach option available in WRF is different for RUC and Noah.

We have not found too much literature about the RUC mosaic option, only the study of Smirnova et al. 2016 (section 3), where they discuss about the advantages of taking into account the sub-grid variability in RUC. After inspection of the code of RUC (below), when the mosaic approach is activated (*mosaic_lu=1*), four variables seem to be computed based on the respective proportion of each land cover category in each pixel: EMIS, ZNT, LAI, PC, which are emissivity, roughness length, leaf area index and a plant coefficient (resistance).

**module_sf_ruclsm.F**

```
!-- mosaic approach to landuse in the grid box
! Use  Mason (1988) Eq.(15) to compute effective ZNT;
!  Lb - blending height =  L/200., where L is the length scale
! of regions with varying Z0 (Lb = 5 if L=1000 m)
     LB = 5.
   if(mosaic_lu == 1) then
   do k = 1,nlcat
    AREA  = AREA + lufrac(k)
    EMISS = EMISS+ LEMITBL(K)*lufrac(k)
    ZNT   = ZNT  + lufrac(k)/ALOG(LB/ZNTtoday(K))**2.
! ZNT1 - weighted average in the grid box, not used, computed for comparison
     ZNT1  = ZNT1 + lufrac(k)*ZNTtoday(K)
     if(.not.rdlai2d) LAI = LAI  + LAItoday(K)*lufrac(k)
     PC    = PC   + PCTBL(K)*lufrac(k)
```

This agrees with what is suggested in Smirnova et al. 2016. These parameters correspond to vegetation ones that depends on the type of vegetation, affecting both the radiation and the dynamics of the flow, but not other soil properties. For the soil properties, another option should be activated (*mosaic_soil=1*), which directly affects soil properties (see code below):

**module_sf_ruclsm.F**

```
! mosaic approach
    if(mosaic_soil == 1 ) then
       do k = 1, nscat
      if(k.ne.14) then
!exclude watrer points from this loop
      AREA   = AREA + soilfrac(k)
      RHOCS  = RHOCS + HC(k)*1.E6*soilfrac(k)
      BCLH   = BCLH + BB(K)*soilfrac(k)
      DQM    = DQM + (MAXSMC(K)-                 &
          DRYSMC(K))*soilfrac(k)
      KSAT   = KSAT + SATDK(K)*soilfrac(k)
      PSIS   = PSIS - SATPSI(K)*soilfrac(k)
      QMIN   = QMIN + DRYSMC(K)*soilfrac(k)
      REF    = REF + REFSMC(K)*soilfrac(k)
      WILT   = WILT + WLTSMC(K)*soilfrac(k)
      QWRTZ  = QWRTZ + QTZ(K)*soilfrac(k)
     Endif
```

On the contrary, the mosaic approach in Noah uses averaged values when the *sf_surface_mosaic* option is set to 1, applying the average for much more parameters: tsk,

qsfc, canwat, snow, tslb, smois, sh2o, hfx, qfx, lh, grdflx, albedo, albbck, emmiss, smbck, znt, zo, lai....). The code is shown below:

**module_sf_noahdrv.F**

```
FAREA = landusef2(i,mosaic_i,j)

TSK_mosaic_avg(i,j) = TSK_mosaic_avg(i,j) +
(EMISS_mosaic(i,mosaic_i,j)*TSK_mosaic(i,mosaic_i,j)**4)*FAREA    ! conserve the
longwave radiation

QSFC_mosaic_avg(i,j) = QSFC_mosaic_avg(i,j) +
QSFC_mosaic(i,mosaic_i,j)*FAREA
CANWAT_mosaic_avg(i,j) = CANWAT_mosaic_avg(i,j) +
CANWAT_mosaic(i,mosaic_i,j)*FAREA
SNOW_mosaic_avg(i,j) = SNOW_mosaic_avg(i,j) +
SNOW_mosaic(i,mosaic_i,j)*FAREA
SNOWH_mosaic_avg(i,j) = SNOWH_mosaic_avg(i,j) +
SNOWH_mosaic(i,mosaic_i,j)*FAREA
SNOWC_mosaic_avg(i,j) = SNOWC_mosaic_avg(i,j) +
SNOWC_mosaic(i,mosaic_i,j)*FAREA

DO NS=1,NSOIL

TSLB_mosaic_avg(i,NS,j)=TSLB_mosaic_avg(i,NS,j) +
TSLB_mosaic(i,NS*mosaic_i,j)*FAREA
SMOIS_mosaic_avg(i,NS,j)=SMOIS_mosaic_avg(i,NS,j) +
SMOIS_mosaic(i,NS*mosaic_i,j)*FAREA
SH2O_mosaic_avg(i,NS,j)=SH2O_mosaic_avg(i,NS,j) +
SH2O_mosaic(i,NS*mosaic_i,j)*FAREA

ENDDO

FAREA_mosaic_avg(i,j)=FAREA_mosaic_avg(i,j)+FAREA
HFX_mosaic_avg(i,j) = HFX_mosaic_avg(i,j) +
HFX_mosaic(i,mosaic_i,j)*FAREA
QFX_mosaic_avg(i,j) = QFX_mosaic_avg(i,j) +
QFX_mosaic(i,mosaic_i,j)*FAREA
LH_mosaic_avg(i,j) = LH_mosaic_avg(i,j) +
LH_mosaic(i,mosaic_i,j)*FAREA

GRDFLX_mosaic_avg(i,j)=GRDFLX_mosaic_avg(i,j)+GRDFLX_mosaic(i,mosaic_i,j)
*FAREA

ALBEDO_mosaic_avg(i,j)=ALBEDO_mosaic_avg(i,j)+ALBEDO_mosaic(i,mosaic_i,j)
*FAREA

ALBBCK_mosaic_avg(i,j)=ALBBCK_mosaic_avg(i,j)+ALBBCK_mosaic(i,mosaic_i,j)*
FAREA

EMISS_mosaic_avg(i,j)=EMISS_mosaic_avg(i,j)+EMISS_mosaic(i,mosaic_i,j)*FAREA

EMBCK_mosaic_avg(i,j)=EMBCK_mosaic_avg(i,j)+EMBCK_mosaic(i,mosaic_i,j)*FA
REA

ZNT_mosaic_avg(i,j)=ZNT_mosaic_avg(i,j)+ALOG(ZNT_mosaic(i,mosaic_i,j))*FARE
A
```

```
Z0_mosaic_avg(i,j)=Z0_mosaic_avg(i,j)+ALOG(Z0_mosaic(i,mosaic_i,j))*FAREA

LAI_mosaic_avg(i,j)=LAI_mosaic_avg(i,j)+LAI_mosaic(i,mosaic_i,j)*FAREA
        if(RC_mosaic(i,mosaic_i,j) .Gt. 0.0) Then
          RC_mosaic_avg(i,j)                                              =
RC_mosaic_avg(i,j)+1.0/RC_mosaic(i,mosaic_i,j)*FAREA
        else
          RC_mosaic_avg(i,j)              =              RC_mosaic_avg(i,j)              +
RC_mosaic(i,mosaic_i,j)*FAREA
        End If
        ENDDO                        ! ENDDO FOR mosaic_i = 1, mosaic_cat
```

In our case, the albedo is the variable which seems to have a larger impact on the fluxes values, affecting the net radiation available at the surface.

We have tried to clarify these lines in the new version of the manuscript (from line 507):

*However, this merging effect was only slightly observed for the case of the MOSAIC experiment in RUC (Fig. 9). This was probably caused by the fact that the mosaic approach used in RUC is applied for different variables than in Noah (see response to RC1). In the case of RUC, only average emissivity, LAI, roughness length and plant resistance are used based on the percentage of each land cover. In RUC, the averaged albedo, which is the parameter that has the highest impact on the net radiation of each grid cell, subsequently affecting SH and Le, is not used. Figure 10 shows the albedo differences between NEW-LC and MOSAIC used by the Noah (upper figures) and RUC models (bottom figures). While the values for MOSAIC-Noah (Fig. 10b) consisted of a weighted average from the different LC of each grid cell, it was not the case for MOSAIC-RUC (Fig. 10d), which diminished the impact of the mosaic approach application in the RUC model.*

7. In the final paragraph of the conclusions, a number of uncertainties in the flux observations are mentioned. It would have been good to address these more formally throughout the paper, but I would not recommend going back to do that now. The paragraph also urges even more comprehensive deployments in the future. Given the size and scope of BLLAST, I think it is unlikely that we will ever see a better-instrumented area. What is needed is better methods of coping with the inevitable limitations of the observations. We will definitely never see comprehensive instrumentation at global scale, so we need to think better about how to do a good job of modeling places we can't measure.

We do agree with your comment, but we think we can always learn from past experiments like BLLAST. For example, in our case it would have been very useful to have some flux measurements in urban and in deciduous forest for this study (although it should be noted that the main objective of BLLAST was not to study the surface heterogeneity specifically but the afternoon transition). In any case, we do agree that having measurements for all the specific covers of each site is a very difficult task, and we should be able to find better ways to extrapolate the current scientific knowledge to estimate fluxes in places where it is difficult to measure. We have removed a too ambitious sentence in the conclusion section from the previous version of the manuscript.

---

## Author Comment (AC3)

**RESPONSES TO REVIEWER #2**

**RC2**: 'Comment on gmd-2020-371', Anonymous Referee #2, 22 Mar 2021

GENERAL COMMENT

The authors analyze the performance of four land surface models (LSMs) in reproducing the sensible and latent heat fluxes. The model performance is assessed using sensible and latent heat fluxes from the second intensive observational period of the BLLAST campaign. The fluxes consist in area averaged values calculated using observations at five different land covers and a high resolution LC classification, CESBIO, derived from Landsat-5 data. Although the derivation of the fluxes may be subjected to considerable uncertainties, the study goes beyond the more traditional comparison against fluxes at one location. The manuscript is well written and provides a clear description of the results with a detailed characterization of the LSMs performance. The evaluation should be useful to refine the LSMs formulation in order to improve the representation of the surface fluxes. The manuscript could be accepted as it is. I provide below a few specific comments, all of the of minor character, that the author should consider to further increase the value of the manuscript.

The authors would like to thank Reviewer #2 for her/his interesting suggestions that have served to better achieve the objectives of the manuscript. Please, find below (in blue) detailed answers to your suggestions.

SPECIFIC COMMENTS

1. In order to isolate the effects of the LSMs the authors can inspect the sensitivity to the initial and boundary conditions. The authors used the NCEP-FNL data to create the initial and boundary conditions and one wonders about the sensitivity of the results to this choice. This is particularly relevant for the initialization of the soil temperature and moisture in WRF. Inspecting the impact of other sources of initial and boundary conditions (e.g. ERA-5) would be a valuable addition.

This is a quite interesting suggestion that we have included in the new version of the manuscript. Besides an additional experiment checking the impact of the spin up in the initial soil moisture values (as suggested by reviewer 1), we have also checked the impact of using a different database for the initial and boundary conditions (ERA-INTERIM, which resolution is more similar to NCEP-FNL than ERA-5). The description of this experiment has been included in the new Section 2.3.5 and the results are included in Section 3.5.

As expected, the results changed, but only slightly. In any case, more experiments in this line (analysing the impact of other data, different horizontal and vertical resolution, etc.) are very interesting to be analysed in a different study.

2. Do the authors have any hypothesis for the different performance of CLM4 in reproducing urban fluxes (Fig. 7)?

Indeed, this LSM showed very large Le values and very low SH, contrary as expected (in Noah the urban class produces almost no evaporation). We do not have a hypothesis for this behaviour and contacting the model developers can help. However, we have decided to remove this LSM in the new version of the manuscript due to some issues found with the simulation of the radiative components.

As commented in the reviewer #1 response, this has been done due to a strange and unrealistic radiative effect found in the inner domain of WRF: after the second analysed day, there is a delay in the morning increase of SWDOWN, showing values equal to 0 W/m$^2$ until some hours after sunrise. This error also appears in subsequent analysed days, but with a delay that lasts 3 hours more every day. This possible bug leads to an unrealistic surface energy budget, affecting the fluxes.

In our case the issue only affected some hours in the morning of day 19$^{th}$ since we only analysed the second simulated day (19$^{th}$ June) from 09 to 15 UTC, but the simulation is not correct. We have tried to solve it and ask the WRF scientific community ( https://wrfforum.com/viewtopic.php?f=43&t=11824 ), but we do not have a solution at the current date. Therefore, we have decided to remove CLM4 from the paper (all the manuscript lines discussing CLM4 and its figures have been removed). Figure R1 shows this issue for a period used in the SPIN-UP simulations:

[Figure]

**Figure R1.** Short wave radiation simulated by Noah and CLM from 17 April 2011 to 22 April 2011 in the central pixel of the simulations for Noah and CLM4, illustrating the issue found in CLM4 (not solved at the present day) and affecting the total radiation available at the surface when several days are simulated. Note how this is observed only in the inner domain (4$^{th}$ domain of 1 km of resolution) and in land pixels (in lake pixels the radiation seems normal). We have tried to change different options in the WRF namelist.input file without success.

3. In order to generalize LSMs performance for the present region, other days should be analyzed to see if results are consistent. This may be well beyond the objectives of the present study, but some discussion in this direction could be added to Section 5.

We do agree completely. We have tried to evidence this even more in lines 676-683 of the new version of the manuscript, being quite interesting for other research focused on analysing different conditions over the same area. In any case, the analysed day (intensive observation period) is representative of the general conditions of the rest of the IOPs, as indicated in lines 114 to 116.

---

## Author Response (AR2)

**12 May 2021**

**RESPONSES TO EDITOR COMMENTS** (Leena Järvi, 11 May 2021)

Thank you for the great work you have made in responding the reviewer comments. There are a few minor points to be addressed before I can accept the manuscript to GMD: 1) Could you add the equations used to calculate the performance scores (RMSE, BIAS and std) in section 2.1.3, and 2) In code and data availability section, the initial and boundary layer data should be cited as suggested on the website (https://rda.ucar.edu/datasets/ds083.2/#metadata/detailed.html?_do=y). Please, correct this. I also had a look on the OSO CESBIO website but as it is written in French it is difficult to see whether there is some more appropriate referencing methods than simply linking the website as this is not the best practice. Could you have a look and think whether there would be more sustainable way to refer the dataset?

We would like to thank the editor for these final suggestions. We have taken them into account in the new submitted article (an additional version with "tracked changes" has also been submitted).

1) We have added the equations of RMSE, BIAS and std in the appropriate section (and slightly changed the associated text).

2) We have updated the citation of the initial and boundary data used (NCEP-FNL). Besides, we have added the citation of the ERA-INTERIM data also used in one of the pre-experiments added after the review process.

   We have asked to the CESBIO team in charge of the land-use dataset, but, unfortunately, they do not have DOI for the 2011 dataset used in our study, only the website is available to be cited (and the article Inglada et al., 2017).

We have corrected some small typo errors in the manuscript, clearly visible in the "tracked changes" version.

We hope that this new version of the manuscript is now ready for publication in GMD.

Best regards,

Carlos Román-Cascón and co-authors.